



# Radiative fluxes in the High Arctic region derived from ground-based lidar measurements onboard drifting buoys

Lilian Loyer[1], Jean-Christophe Raut[1], Claudia Di Biagio[2], Julia Maillard[1], Vincent Mariage[1], and Jacques Pelon[1]

[1]LATMOS/IPSL, Sorbonne Université, Université Versailles Saint Quentin, CNRS, Paris, France
[2]Université de Paris and Univ Paris Est Creteil, CNRS, LISA, F-75013 Paris, France

**Correspondence:** Lilian Loyer (lilian.loyer@latmos.ipsl.fr)

**Abstract.**

The Arctic is facing drastic climate changes that are not correctly represented by state-of-the-art models because of complex feedbacks between radiation, clouds and sea-ice surfaces. A better understanding of the surface energy budget requires radiative measurements that are limited in time and space in the High Arctic (>80°N) and mostly obtained through specific expeditions.

Six years of lidar observations onboard buoys drifting in the Arctic Ocean above 83°N have been carried out as part of the IAOOS (Ice Atmosphere arctic Ocean Operating System) project. The objective of this study is to investigate the possibility to extent the IAOOS dataset to provide estimates of the shortwave (SW) and longwave (LW) surface irradiances from lidar measurements on drifting buoys. Our approach relies on the use of the *STREAMER* radiative transfer model to estimate the downwelling SW scattered radiances from the background noise measured by lidar. Those radiances are then used to derive

estimates of the cloud optical depths. In turn, the knowledge of the cloud optical depth enables to estimate the SW and LW (using additional IAOOS measured information) downwelling irradiances at the surface. The method was applied to the IAOOS buoy measurements in spring 2015, and retrieved cloud optical depths were compared to those derived from radiative irradiances measured during the N-ICE (Norwegian Young Sea Ice Experiment) campaign at the meteorological station, in the vicinity of the drifting buoys. Retrieved and measured SW and LW irradiances were then compared. Results showed overall

good agreement. Cloud optical depths were estimated with a rather large dispersion of about $47\%$. LW irradiances showed a fairly small dispersion (within $5\ \mathrm{W\,m^{-2}}$), with a corrigible residual bias ($3\ \mathrm{W\,m^{-2}}$). The estimated uncertainty of the SW irradiances was $4\%$. But, as for the cloud optical depth, the SW irradiances showed the occurrence of a few outliers, that may be due to a short lidar sequence acquisition time (no more than four times $10\ \mathrm{mn}$ per day), possibly not long enough to smooth out cloud heterogeneity. The net SW and LW irradiances are retrieved within $13\ \mathrm{W\,m^{-2}}$.

## 1 Introduction

The Arctic is facing rapid climate changes with surface temperatures increasing twice as fast then the rest of the world (Serreze and Barry, 2011). One of the most notable changes in the Arctic region over the recent decades is the reduced sea ice coverage and thickness (Meier et al., 2014). This thinner ice breaks more easily to form open water leads, increasing the heat flux





and warming the surface through enhanced absorption of SW radiation, and the consequent ice albedo feedback (Semmler
et al., 2012). Not only does this change affect the Arctic climate, but also has an impact on the global temperature trend and
mid-latitude climate (Huang et al., 2017; Vihma, 2014).

Arctic clouds, which crucially contribute to regulate the surface radiative budget, are still poorly represented in regional
and global models (Lacour et al., 2018; Jung et al., 01 Sep. 2016; Taylor et al., 2019), and the thermodynamical and radiative
feedbacks between ice surfaces, the boundary layer and clouds are still not satisfactorily understood (Gierens et al., 2020;
Huang et al., 2019; Tan and Storelvmo, 2019; Loewe et al., 2017). In the Arctic, clouds cover up to 80% of the region at
all time and are primarily composed of low-level mixed phase clouds (Ruiz-Donoso et al., 2020; Mioche et al., 2017; Intrieri,
2002). With the increase in the number of open water leads and the associated enhanced heat and humidity fluxes, the number of
clouds may also be enhanced in the Arctic (Abe et al., 2016; Palm et al., 2010). Palm et al. (2010); Li et al. (2020) nevertheless
suggested that the typical mixed low-level clouds observed in the Arctic could be replaced by high-level ice clouds as a result
of the increased heat flux. This would deeply change the thermodynamical and radiative relationship between clouds and sea
ice.

The underdetermined knowledge on the thermodynamical and radiative feedbacks of clouds also impacts the understanding
of the surface energy budget in the Arctic (Taylor et al., 2019). Clouds together with surface properties (spectral albedo,
emissivity, temperature) indeed play a major role on the radiative budget, which is the primary source in the surface energy
budget in the Arctic region (Serreze et al., 2007). The presence of clouds tends to decrease the shortwave (SW) irradiance at the
surface through the scattering of the incoming solar radiation. This effect may vary between $-50\ \mathrm{W\,m^{-2}}$ to zero depending
on the surface albedo, the solar zenith angle and the cloud liquid water content (LWC) (Sedlar et al., 2010). On the other
hand, the presence of clouds increases the longwave (LW) irradiance between $65$ and $85\ \mathrm{W\,m^{-2}}$ by absorbing the terrestrial
radiation and re-emitting it back to the surface (Sedlar et al., 2010). Ebell et al. (2020) characterized the cloud radiative effect
at Ny-Ålesund, Svalbard, in Norway and found an annual surface warming effect by clouds of $11.1\ \mathrm{W\,m^{-2}}$ in agreement with
the results of Sedlar et al. (2010).

A pronounced seasonal variability in the radiative fluxes is observed in the Arctic (Uttal et al., 2002; Perovich, 2002; Walden
et al., 2017). In winter the radiative budget is negative as the upwelling LW irradiance is larger than the downwelling flux
(Walden et al., 2017; Perovich, 2002) and SW irradiance is absent. The SW irradiance increases as the solar zenith angle
decreases and compensates the net LW cooling in spring (Walden et al., 2017). The SW irradiance is higher in summer, leading
to a positive surface radiative budget during that season (Walden et al., 2017). A seasonal variability of the LWC in mixed-phase
clouds is also observed, with the lowest values in winter and the highest in summer (Intrieri, 2002; Taylor et al., 2019; Ebell
et al., 2020). The radiative budget is therefore modulated as mixed-phase clouds with a higher LWC, especially in summer,
are optically thicker (Bennartz et al., 2013; Zhang et al., 2019). The surface cloud radiative effect is therefore positive from
September to April-May and negative in summer (Walden et al., 2017; Ebell et al., 2020).

Observing how the radiative and surface energy budget evolves in the changing Arctic environment is key to understand
the future of this region (Sedlar et al., 2010). And there is a crucial need for observational data (Blanchard et al., 2014; Liu
et al., 2017). Multiple large-scale expeditions have taken place in the Arctic to study the surface energy budget, like the Surface



Heat Balance of the Arctic (SHEBA) in 1998 (Uttal et al., 2002), the Arctic Summer Cloud Ocean Study (ASCOS) in 2014
(Tjernström et al., 2014), the Norwegian Young Sea Ice Experiment (N-ICE) in 2015 (Cohen et al., 2017), the Arctic Clouds in
Summer Experiment (ACSE) (Sotiropoulou et al., 2016), the Arctic CLoud Observations Using airborne measurements during
polar Day (ACLOUD) aircraft campaign and the Physical feedbacks of Arctic boundary layer, Sea ice, Cloud and AerosoL
(PASCAL) ice breaker expedition (Wendisch et al., 01 May. 2019), and recently the Multidisciplinary drifting Observatory for
the Study of Arctic Climate (MOSAiC) in 2020 (https://www.mosaic-expedition.org). During those campaigns, crucial data
have been sampled, improving our understanding of the Arctic region and of the complex interplay between radiation, clouds
and surface properties, but such expeditions are limited in time and cover only a limited part of the Arctic region. Satellite-
based remote sensing instruments documenting clouds, such as CALIOP aboard CALIPSO (Lacour et al., 2017) or CloudSat
(Stephens et al., 2018) are limited to latitudes below $82°$N because of the satellite flight path (Winker et al., 2009). To respond to
the need for more observations in the Arctic and compensate the lack of satellite observations at the highest latitudes, multiple
buoys have been deployed in the Arctic Ocean between 2014 and 2019 as part of the IAOOS (Ice Atmosphere arctic Ocean
Operating System) project (Mariage et al., 2017; Maillard et al., 2021). Buoys have also limitations, e.g. their limited available
space and the capacity of their batteries, but spread across the Arctic Ocean allows a wider spatial coverage and longer periods
of observations, where other remote sensing instruments (for example onboard satellites) are blind.

This study investigates the possibility to provide a more complete dataset from the IAOOS observations, including estimated
radiative fluxes from buoys lidar data that do not have such measurements. The paper presents a method to estimate the
cloud optical depths and the SW and LW irradiances using a combination of zenith-pointing backscatter lidar measurements
and radiative transfer model outputs. A similar approach has already been used by Chiu et al. (2014) on warm mid-latitude
continental low-level clouds. They retrieved the cloud optical depth from ground-based lidar solar background measurements,
located in ARM Oklahoma, from 2005 to 2007. Here, a similar methodology is used on cold low-level mixed-phase clouds
in the Arctic Ocean to derive both optical depths and radiative irradiances. Our approach is applied to observations sampled
from buoys as part of the IAOOS project, and its limits and associated uncertainties are discussed. The results are compared
to SW and LW irradiances measurements at the vicinity of the buoy from the N-ICE expedition. Section 2 details the dataset
used in this study. The radiative transfer model and the approach developed in this study are described in Sect. 3. In Sect. 4, the
methodology is applied to lidar data aboard buoys from the IAOOS campaigns, uncertainties and limitations of the approach
are discussed and results are compared to standard measurements of radiative fluxes. Conclusions are finally drawn (Sect. 5).

## 2 Observational dataset

### 2.1 IAOOS lidar data aboard buoys

The main objective of the IAOOS project, led by Sorbonne University, was to collect real time observations of the ocean,
ice, snow and atmosphere of the Arctic simultaneously. The project allowed several field experiments between 2014 and 2019
using autonomous buoys equipped with atmospheric and oceanic profiling instruments and distributed over the Central Arctic
region (Provost et al., 2015). Each floating plaform was locked into the pack ice during several months, left to drift with it



and tacked by GPS every day. The buoys were equipped with a micro backscatter lidar shooting in the zenith direction in the near infrared, designed for cloud and aerosol profiling in the lower troposphere (Mariage et al., 2017), surface meteorological sensors on a 502 luft weather station at 2 m, snow and ice temperature profiler (Koenig et al., 2016) and ocean profilers (Provost et al., 2015). Figure 1 presents the tracks of those autonomous platforms and the N-ICE expedition camp site during the field campaign.

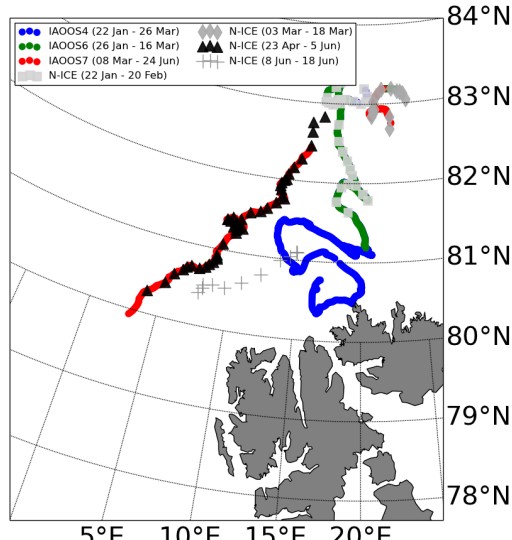

**Figure 1.** Map of the IAOOS buoys tracks (colored dotted lines) and the trajectories of the different legs followed by the research vessel Lance during the N-ICE field experiment (gray and black symbols) in the North of Svalbard in 2015. In this study, the data from the IAOOS7 buoy (red dotted line) and the leg 3 of N-ICE (triangles) in the springtime period are used.

This study will mostly focus on the lidar measurements acquired from January to June 2015 onboard three buoys during the N-ICE field experiment. They have already been used to investigate the occurrence and distribution of winter to summertime clouds and aerosols in the high Arctic Ocean (Di Biagio et al., 2018, 2020; Maillard et al., 2021). The lidar emits a laser at a wavelength of $808\,\mathrm{nm}$ and the field of view of the receiver is $\sim 1.4 \times 10^{-6}$ sr. Measurements were performed 2 to 4 times a day with a 10-minute averaging sequence for each profile. During that interval, the window topping the lidar emitter and receiver was heated to avoid frost deposition. Unfortunately, in wintertime (from mid-December to early March), when temperatures could reach down to 230K, the window heating system was not sufficient, leading to a layer of frost on the top of it that strongly attenuated the lidar signal or even blinded it. As a consequence, data discussed in this study only cover the spring period from April to June 2015 and profiles with attenuated signal by frost (frost index lower than 0.7 as defined by Mariage (2015)) were removed. This corresponds to 20 profiles out of 65 for the 2015 spring period. The approach proposed in this paper relies on





the measured solar background $B$ that is calculated as the average raw lidar signal above 20 km, where photons are assumed
to be emitted by the sun only, as will be explained in Sect. 3.

## 2.2 N-ICE2015 radiative measurements

As part of the N-ICE campaign that took place from January to June 2015, the research vessel Lance drifted with several floes
(Walden et al., 2017; Cohen et al., 2017), two in wintertime (from January to March) and two from late spring to early summer
(April to June) (Fig. 1). Three IAOOS buoys were deployed during this campaign in the northern part of the Nansen basin and
drifted in the first three ice floes close to the research vessel. No buoy was deployed during the last floe as it drifted closer to the
ice edge (Granskog et al., 2018). Atmospheric observations were mostly performed at a ice camp about 300 m away from the
Lance vessel (Kayser et al., 2017; Granskog et al., 2018). Vertical profiles of temperature, relative humidity, and winds were
reported twice a day from radiosondes from 16 January to 26 March, and from 17 April to 22 June (Hudson et al., 2017). In
particular, the temperature at two meters, measured with a ventilated and shielded Vaisala HMP-155A sensor, had an accuracy
of $0.3°C$ (2.4%) (Graham et al., 2017; Cohen et al., 2017).

Radiative (up and down) fluxes at the surface ($\sim 1$ m above snow) were measured on all four floes (Walden et al., 2017).
Downwelling ($F_{SW}^{\downarrow}$, $F_{LW}^{\downarrow}$) and upwelling ($F_{SW}^{\uparrow}$, $F_{LW}^{\uparrow}$) SW and LW irradiances were measured at 1 min resolution with Kipp &
Zonen CMP22 and CGR4 radiometers (having a 200 to 3600 nm and a 4.5 to 42 μm bandwidths, respectively). Their accuracy
was 5 W m$^{-2}$ ($\sim 3\%$) for the SW and 3 W m$^{-2}$ ($\sim 2\%$) for the LW (Walden et al., 2017; Hudson et al., 2016). According
to the quality flags (QF) classification introduced by Walden et al. (2017), we only considered observations defined as "good
data" (QF=0) and selected only days having more than 70% of data with QF=0. Finally, we also eliminated 41 days when the
IAOOS buoy (IAOOS7) was too far from the ice camp. This removed a total of 63 days of measurements. The radiative fluxes
observed in the spring period (April to June) and used as a reference in this study are presented in Fig. 2.

During the drift, the solar zenith angle $\theta$ decreases from 66 $°$ in April to 58 $°$ in June at noon, with a diurnal variation of $\sim 20$
$°$. The SW irradiances $F_{SW}^{\downarrow}$ and $F_{SW}^{\uparrow}$ present a clear diurnal cycle with minimal absolute values reaching $\sim 80$ W m$^{-2}$ and
$\sim 67$ W m$^{-2}$, respectively (Fig. 2). In springtime, Walden et al. (2017); Cohen et al. (2017) indicated that the meteorological
conditions were quite stable and that the surface temperature did not significantly vary. The resulting $F_{LW}^{\uparrow}$ was very stable
($F_{LW}^{\uparrow} \simeq -277 \pm 21$ W m$^{-2}$) during the whole period. As a consequence, the observed variations in $F_{LW}^{\downarrow}$ are mostly from
changes in the cloud cover or properties. Both the SW and LW irradiances are affected by clouds, and there were only a few
cases in the analysed period when clear sky conditions were met. On 23 May, the clear sky conditions encountered during
24 hours led to extreme values of $F_{SW}^{\downarrow} \simeq 550$ W m$^{-2}$ (maximum) and $F_{LW}^{\downarrow} \simeq 160$ W m$^{-2}$ (minimum) on 1 minute resolution
135 (Walden et al., 2017).

## 2.3 ERA5 reanalyses

The temperature $T_c$ of the cloud at the effective altitude $z_c$ of its emission (Sect. 3.3) is determined from the temperature profile
obtained from the European Centre for Medium-Range Weather Forecasts Reanalyses v5 (ERA5) (Hersbach et al., 2018). The
ERA5 data have a resolution of $0.25° \times 0.25°$ and are interpolated at the buoy location. Figure 3 shows a good agreement





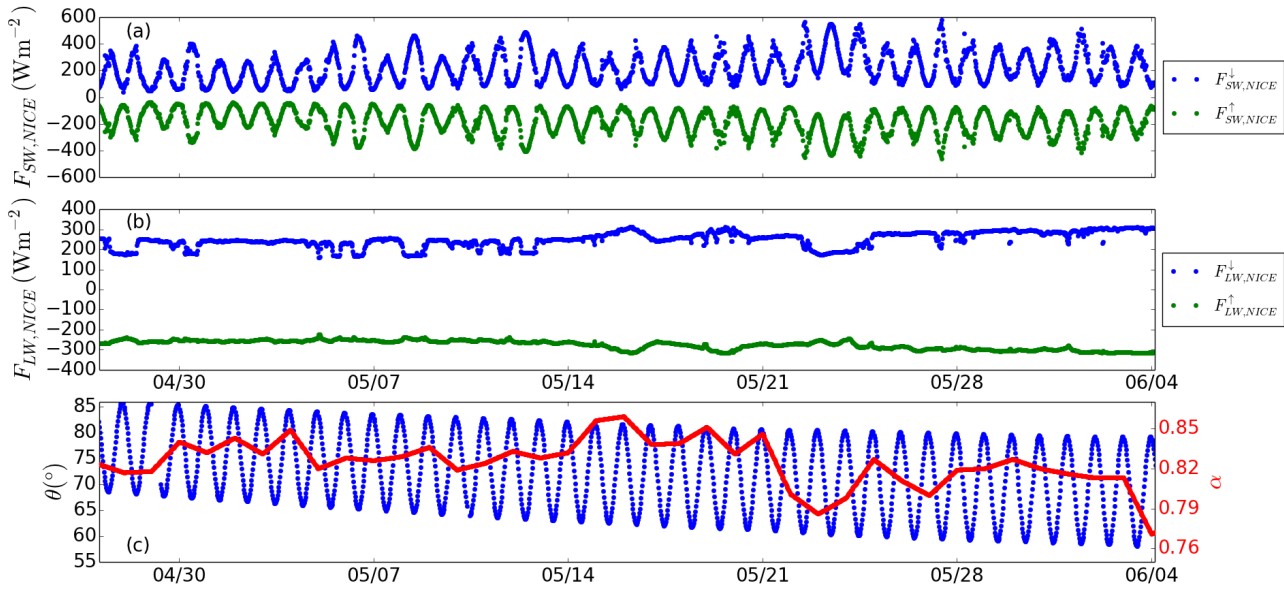

**Figure 2.** SW (a) and LW (b) irradiances measured at 1-min resolution, solar zenith angle (c, blue) and surface albedo (c, red) reported as the mean around noon during the N-ICE campaign in 2015 North of the Svalbard in springtime (April-June). Upwelling and downwelling irradiances are represented in green and blue, respectively.

between the temperature $T_c$ obtained from ERA5 reanalyses and measured by radiosoundings during the N-ICE expedition, with Pearson correlation coefficient of $0.95$. The ERA5 temperature therefore tends to be a good estimate of $T_c$, even though the simulated values are slightly colder than the N-ICE measurements, a cold bias of $-0.3$ K and a root mean square error (RMSE) of $1.6$ K.

## 3  Method

### 3.1  Radiative Transfer Model : *STREAMER*

In this study, the SW (direct and scattered) radiances and irradiances were calculated with the *STREAMER*-Version 3 radiative transfer model (Key and Schweiger, 1998) for a wide variety of atmospheric and surface conditions. Calculations are based on the plane-parallel theory of radiative transfer with a discrete ordinate (DISORT) solver. The source code was modified to output the scattered radiance simultaneously with the SW irradiance at the surface. The model was set to have a field of view similar to that of the receiver of the IAOOS lidar. Upwelling and downwelling irradiances were computed over 24 SW bands in the same bandwidth as the one of the pyranometer used during the N-ICE experiment (200 to 3600 nm). We chose





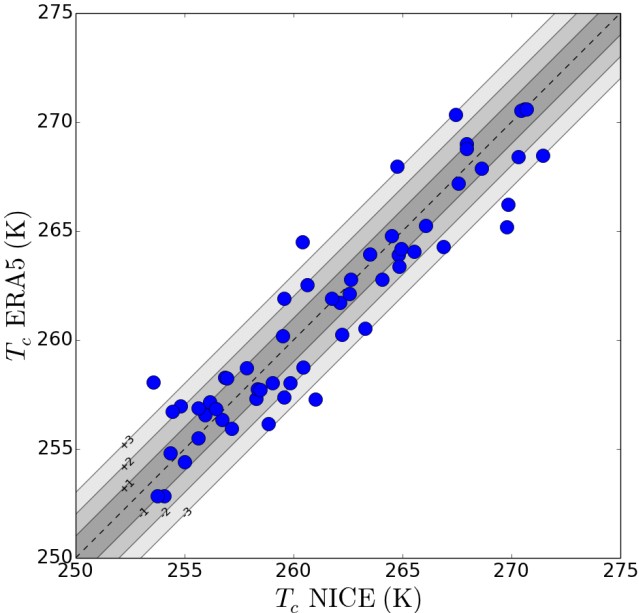

**Figure 3.** Comparison of the temperature $T_c$ at effective cloud height $z_c$ obtained from ERA5 reanalyses and measured by radiosoundings during the N-ICE campaign in spring 2015. The dotted line present a 1:1 ratio. The shaded areas show $\pm 1$ K, $\pm 2$ K and $\pm 3$ K differences from the 1:1 ratio.

snow-covered surface conditions with a visible spectral albedo varying from $0.73$ to $0.87$. It is in agreement with the values reported by Merkouriadi et al. (2017); Granskog et al. (2018) during the N-ICE campaign, suggesting that the field of view of the upwelling instruments always encompassed a snow-covered sea-ice surface. The vertical profiles of pressure, temperature and humidity were interpolated from the profiles measured by radiosoundings (Hudson et al., 2017) during the N-ICE field experiment and averaged over spring 2015. Because the Arctic region is typically covered by mixed-phase clouds, representing $80\%$ of the cloud coverage (McFarquhar et al., 2007; Shupe et al., 2011; Morrison et al., 2011), we assumed the presence of mixed-phase clouds in our *STREAMER* simulations with a cloud optical depth ($\tau$) varying from 0 (clear sky conditions) to 200 (opaque cloud). Clouds were assumed to have a fixed geometrical depth with base and top at 200 and 800 m above mean sea level, respectively, and to be composed of two cloud layers: a 100 m-width cloud layer composed of $6.9 \pm 1.8$ µm-diameter water droplets overcoming a 500 m-width cloud layer composed of $25.2 \pm 3.9$ µm-diameter hexagonal ice crystals, as described by McFarquhar et al. (2007).

Figure 4 shows the total downwelling SW irradiance $F^{\downarrow}_{\mathrm{SW}}$ and the scattered downwelling SW radiance $L^{\downarrow}_s$ computed from the *STREAMER* radiative transfer model as a function of the solar zenith angle $\theta$ changing from $50°$N to $90°$N and of the cloud optical depth $\tau$ varying 0 (clear sky conditions) to 200 (opaque cloud) for an albedo $\alpha$ of $0.82$. The scattered downwelling SW irradiance $F^{\downarrow}_{\mathrm{SW,s}}$ is also computed from *STREAMER* but is not shown here. $F^{\downarrow}_{\mathrm{SW}}$ monotonically decreases with increasing



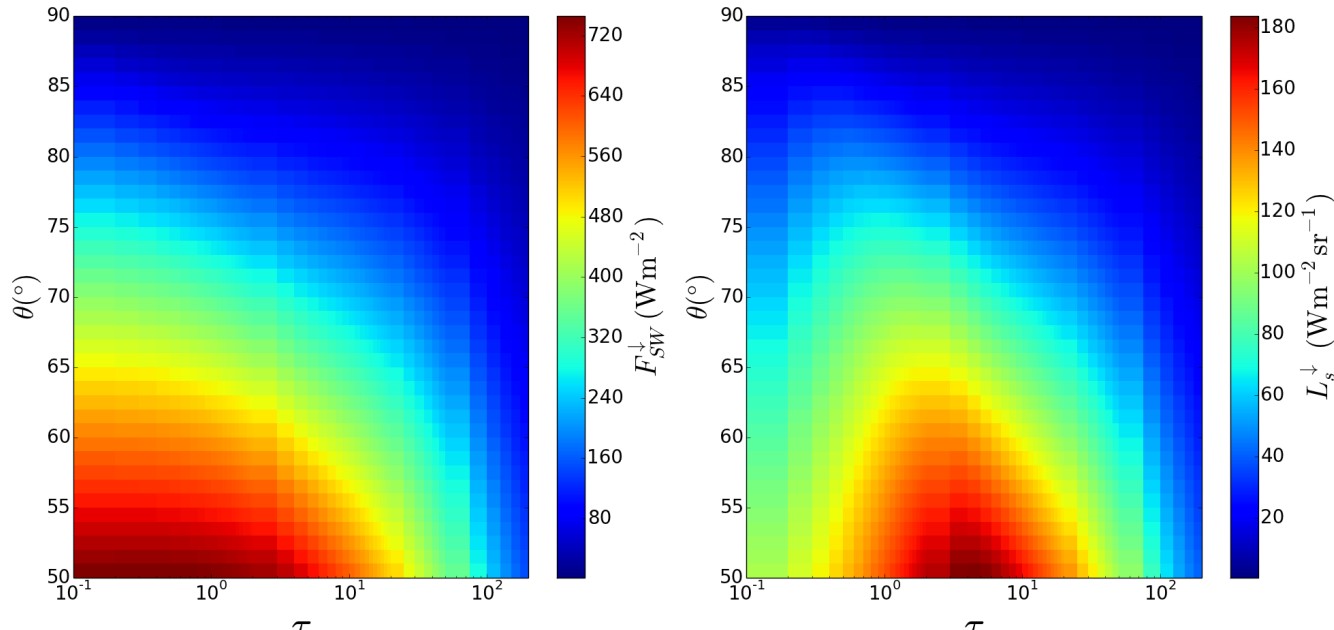

**Figure 4.** Total downwelling SW irradiance $F_{\mathrm{SW}}^{\downarrow}$ (left) and scattered downwelling SW radiance $L_s^{\downarrow}$ (right) computed from the *STREAMER* radiative transfer model as a function of the solar zenith angle $\theta$ changing from 50 °N to 90°N and of the cloud optical depth $\tau$ varying 0 (clear sky conditions) to 200 (opaque cloud). Clouds are assumed to have a fixed geometrical depth between 200 and 800 m above mean sea level and to be composed of two cloud layers: a 100 m-width cloud layer composed of $6.9 \pm 1.8$ μm-diameter water droplets overcoming a 500 m-width cloud layer composed of $25.2 \pm 3.9$ μm-diameter hexagonal ice crystals, as described by McFarquhar et al. (2007). The surface is supposed covered by snow with a albedo of 0.82 in the visible.

values of $\theta$ as the sun position is lower over the horizon, and increasing values of $\tau$ as the cloud becomes more opaque. $L_s^{\downarrow}$ also becomes weaker when the sun goes down above the horizon. In contrast, $L_s^{\downarrow}$ increases with $\tau$ as long as $\tau$ remains lower than a threshold value $\tau_{\max}(\theta)$, between 1 and 4, depending on the solar zenith angle, and decreases afterwards.

## 3.2 Estimation of the SW irradiance from lidar measurements

The total downwelling SW irradiance $F_{\mathrm{SW}}^{\downarrow}$ can be written as the sum of its direct $F_{\mathrm{SW,d}}^{\downarrow}$ and scattered $F_{\mathrm{SW,s}}^{\downarrow}$ components, as a function of the cloud optical depth $\tau$ :

$$F_{\mathrm{SW}}^{\downarrow}(\tau,\theta,\alpha) = F_{\mathrm{SW,d}}^{\downarrow}(\tau,\theta) + F_{\mathrm{SW,s}}^{\downarrow}(\tau,\theta,\alpha) \tag{1}$$

The direct component is calculated as :

$$F_{\mathrm{SW,d}}^{\downarrow}(\tau,\theta) = S_0 C \cos(\theta) e^{-(\tau+\tau_r)/\cos(\theta)} \tag{2}$$





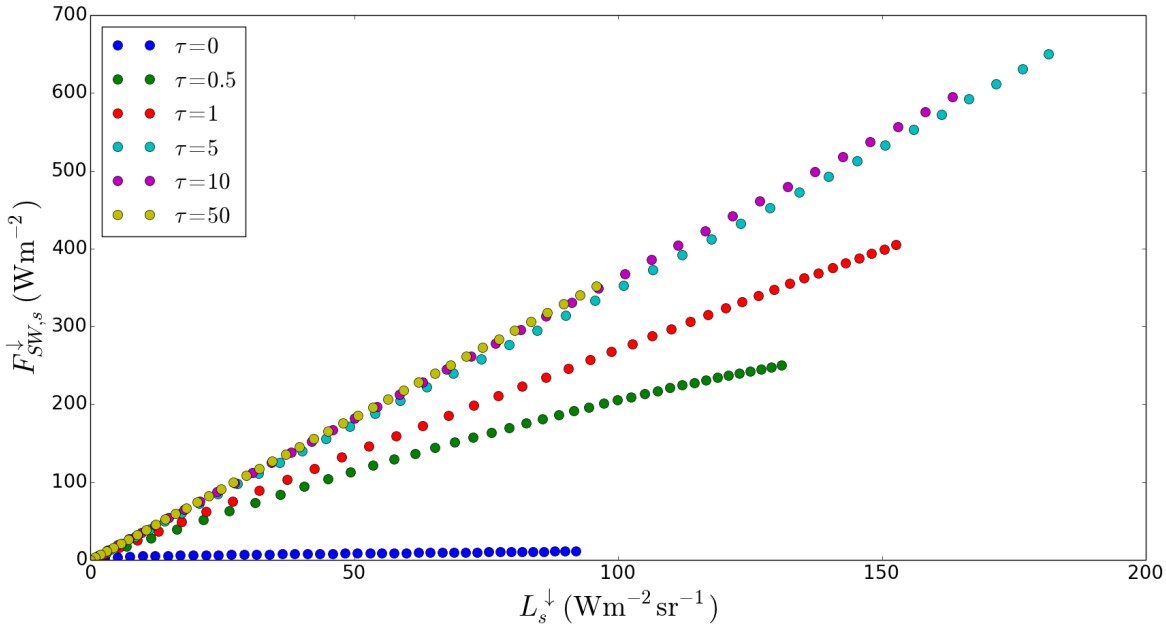

**Figure 5.** Comparison of the downwelling scattered SW irradiances and the scattered downwelling SW radiances for 6 values of $\tau$ (0, 0.5, 1, 5, 10, 50) and various values of $\theta$ (from $50°$ to $90°$) using *STREAMER* with $\alpha = 0.82$.

where $S_0 = 1354.2\,\mathrm{W\,m^{-2}}$, $C$ and $\tau_r$ are the solar constant, the Sun-Earth distance correction factor, and the Rayleigh optical depth of the atmosphere, respectively. $\tau_r$ is calculated using the vertical profiles of temperature and pressure measured during the N-ICE campaign according to Bucholtz (1995). Similarly, the upwelling SW irradiance is calculated as the reflected SW irradiance by the surface :

$$F_{\mathrm{SW}}^{\uparrow}(\tau, \theta, \alpha) = \alpha F_{\mathrm{SW}}^{\downarrow}(\tau, \theta, \alpha) \qquad (3)$$

with $\alpha$ the surface albedo. When the values of $\tau$ and $\theta$ are known, the scattered downwelling SW irradiance $F_{\mathrm{SW,s}}^{\downarrow}$ is calculated from *STREAMER* by integrating the downwelling scattered SW radiance $L_s^{\downarrow}$ over the solid angle of the upwelling hemisphere (fig.5). Platt et al. (1998) showed that the measured solar background $B$ obtained from a lidar backscatter signal is proportional to $L_s^{\downarrow}$ integrated over the solid angle described the field of view of the lidar receptor :

$$B = K_L L_s^{\downarrow} \qquad (4)$$

The constant $K_L$ in $\mathrm{W^{-1}\,m^2\,sr}$ only depends on the instrumental characteristics of the lidar system (Zhang et al., 2018; Mariage et al., 2017; Platt et al., 1998).

Figure 5 presents a comparison between the downwelling scattered SW irradiance and radiance derived from *STREAMER* for 6 values of $\tau$. For each value of $\tau$ (0, 0.5, 1, 5, 10, 50), a quasi-linear relationship between the $F_{\mathrm{SW,s}}^{\downarrow}$ and $L_s^{\downarrow}$ is observed. As a





result, the knowledge of the measured solar background $B$ derived from the lidar helps to determine the downwelling scattered SW radiance using Eq. 4 and the corresponding downwelling scattered SW irradiance from the *STREAMER* simulation (Fig. 5). This leads in turn to a determination of the total SW irradiances $F_{\text{SW}}^{\downarrow}$ and $F_{\text{SW}}^{\uparrow}$ for each value of the solar zenith angle (Eq. 1 and Eq. 3). This approach therefore relies on the knowledge of the cloud optical depth $\tau$ and the $K_L$ constant.

### 3.3  Estimation of the LW irradiance from lidar measurements

The estimation of the downwelling LW irradiance $F_{\text{LW}}^{\downarrow}$ is simply obtained from the equation detailed by Minnis et al. (1993):

$$F_{\text{LW}}^{\downarrow} = c\epsilon_c\sigma T_c^4 + c\epsilon_s\sigma T_s^4(1-\epsilon_c) + (1-c)\epsilon_a\sigma T_{2\text{m}}^4 \tag{5}$$

$\epsilon_s = 0.98$ and $\epsilon_a = 0.7$ are typical emissivities of the surface and the atmosphere, respectively, representative of the Arctic region (Mariage et al., 2017). $T_s$ is the skin temperature of the surface, $T_{2\text{m}}$ the temperature at 2 m and $T_c$ the effective temperature of the cloud. Because the approach used here must be independent of N-ICE measurements and as $T_s$ and $T_{2\text{m}}$

were not measured in spring 2015 on the IAOOS platforms, values of $T_s$ and $T_{2\text{m}}$ are then obtained from ERA5 reanalyses. In the absence of cloud (cloud mask $c = 0$), the downwelling LW irradiance $F_{\text{LW}}^{\downarrow}$ is simply calculated as the downwelling emission flux from the lowest atmospheric layer. In the presence of a cloud (cloud mask $c = 1$), $F_{\text{LW}}^{\downarrow}$ is the sum of two terms: the LW emission of the cloud toward the surface and the reflection by the cloud of the upwelling LW irradiance from the surface $F_{\text{LW}}^{\uparrow}$, given by :

$$F_{\text{LW}}^{\uparrow} = \epsilon_s\sigma T_s^4 \tag{6}$$

The cloud emissivity $\epsilon_c$ is assessed as in Minnis et al. (1993):

$$\epsilon_c = 1 - \exp\left(\frac{-\beta_1\tau^{\beta_2}}{\cos(\theta)}\right) \tag{7}$$

with $\beta_1 = 0.4734$ and $\beta_2 = 1.0216$. When the cloud optical depth $\tau$ is above 3, the cloud is considered to act as a black body, and $\epsilon_c = 1$. The vertical profiles of temperature obtained from ERA5 reanalyses help to interpolate $T_c$ at $z_c$, the effective

altitude of the contributed radiation of the cloud. That latter is estimated from the attenuated scattering ratio profiles in the infrared ($SR_{att,\text{IR}}$). The attenuated scattering ratio $SR_{att}$ derived from the lidar in the visible (808 nm) at range $z$ from the lidar emitter can be expressed as :

$$SR_{att}(z) = \frac{\beta_{att}(z)}{\beta_{mol}(z)} \tag{8}$$

with $\beta_{mol}$ the molecular backscatter coefficient and $\beta_{att}$ the attenuated total backscatter coefficient, linked to the backscatter

coefficient $\beta(z)$ :

$$\beta_{att}(z) = \beta(z)\exp\left(-2\eta S_c \int_0^z \beta(r)\mathrm{d}r\right) \tag{9}$$





Previous analyses during the IAOOS campaign used a lidar ratio (extinction-to-backscatter ratio) of $S_c = 18$ sr and a multiple scattering factor $\eta = 0.8$ (Mariage et al., 2017). Garnier et al. (2012) estimated the ratio between the cloud optical depths in the visible and infrared domains to be roughly 2. $SR_{att,\mathrm{IR}}$ can be therefore be calculated as follows:

$$SR_{att,\mathrm{IR}}(z) = SR_{att}(z)e^{+\eta S_c \int_0^z \beta(r)\mathrm{d}r} \tag{10}$$

$\beta$ is obtained from Eq.9 using a forward inversion (Klett, 1985). In our study, clouds are observed at low altitudes, below 2 km (Fig. 6) and could be co-localized with aerosol layers (Di Biagio et al., 2018). A minimal threshold of 5 on $SR_{att,\mathrm{IR}}$ is therefore used to select only cloud layers. The base and top heights of a specific cloud layer are respectively identified as the altitude where $SR_{att,\mathrm{IR}} \geq 5$ for the first time and the altitude where $SR_{att,\mathrm{IR}} < 5$ above the base height for the first time, if

any. Finally, $z_c$ is determined as the height between the base and the top with the maximum value of $SR_{att}$. This approach is used in all cloud layers identified in a lidar vertical profile, similarly to what Mariage et al. (2017) have already done for the IAOOS measurements in 2014.

## 4    Results and Discussion

In this section, we present the results of the determination of the cloud optical depth $\tau$ and the $K_L$ constant required to assess

the radiative fluxes from the lidar measurements. The estimation of the SW and LW irradiances is then given and compared to values measured during the N-ICE experiment. The limits of the approach are finally discussed.

### 4.1    Determination of the cloud optical depth from the radiometers during N-ICE : $\tau_{\mathrm{NICE}}$

Figure 4 summarizes the values of $F_{\mathrm{SW}}^{\downarrow}$ and $L_s^{\downarrow}$ for various values of $\theta$ and $\tau$ at $\alpha = 0.82$. Determining the cloud optical depth from the radiative fluxes measured by the radiometers during the N-ICE campaign is relatively straightforward. Since

the relationship between $F_{\mathrm{SW}}^{\downarrow}$ and $\tau$ is bijective at a fixed value of $\theta$, $\tau$ can be obtained at any instant (or any solar zenith angle value) in springtime by minimizing the absolute differences between the observed and the simulated values of $F_{\mathrm{SW}}^{\downarrow}$. This determination is referred to as $\tau_{\mathrm{NICE}}$ in the following. For each day, the simulated values of $F_{\mathrm{SW}}^{\downarrow}$ derived from *STREAMER* have been corrected of the Earth-Sun distance. Figure 6 shows the time series of $\tau_{\mathrm{NICE}}$ in spring 2015. The temporal evolution of $\tau_{\mathrm{NICE}}$ confirms that low-level clouds with a large optical depth are very frequent north of Svalbard in spring, as it was discussed

by Di Biagio et al. (2020). Their average optical depth is found to be $16 \pm 10$ and is in agreement with values derived from the ERA5 reanalyses and CERES (Clouds and the Earth's Radiant Energy System) satellite (Di Biagio et al., 2020). On 23 May, clear sky conditions were observed during the whole day (Cohen et al., 2017). The fact that the optical depth $\tau_{\mathrm{NICE}}$ is weak ($< 0.2$) can be explained by the presence of aerosols.

### 4.2    Relation between the solar background $B$ and the downwelling scattered SW radiance $L_s^{\downarrow}$

Assessing the cloud optical depth from the IAOOS lidar measurements is more tricky as it requires a series of steps. The first of them is the calculation of the $K_L$ constant. This coefficient is the theoretical slope of the linear dependency of the solar





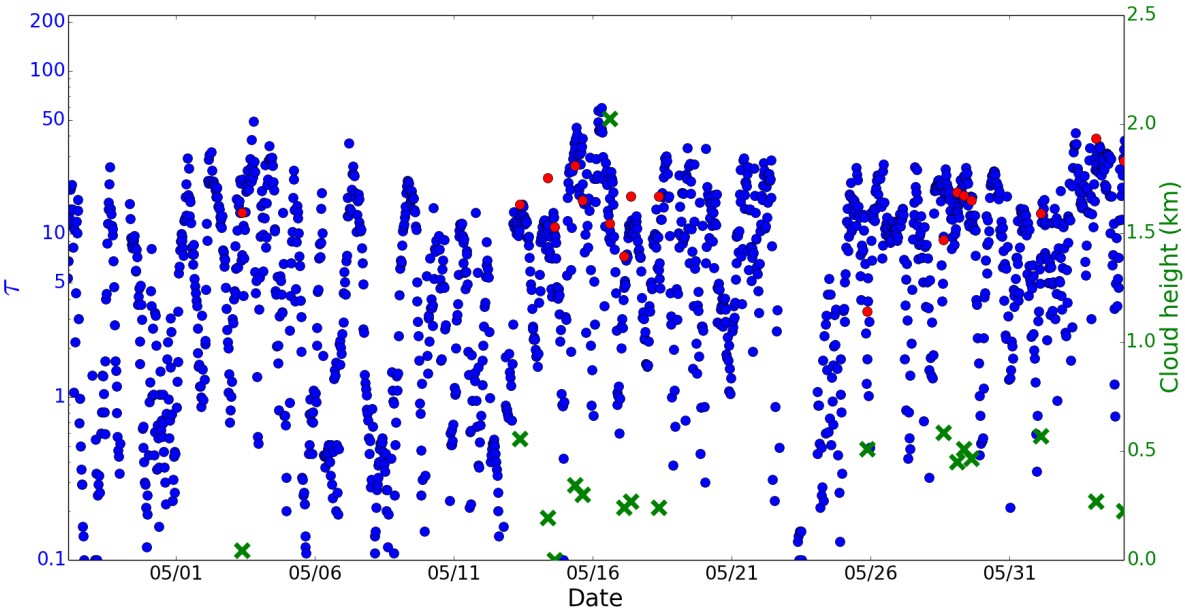

**Figure 6.** Time series of $\tau_{\mathrm{NICE}}$ (blue dots), $\tau_{\mathrm{IAOOS}}$ (red dots) during the N-ICE field expedition in spring 2015. The effective cloud height $z_c$ is represented by green crosses.

background $B$ on $L_s^{\downarrow}$ (Sect. 3.2), and is a sole function of instrumental and optical properties of the lidar system. An accurate knowledge of the instrumental properties of an autonomous lidar in the Arctic Ocean is however a challenge, not only because there is a limited number of clear-sky days enabling to check the lidar calibration, but also as a frost layer is often deposited
on the window of the lidar, disturbing the received signal (Sect. 2.1). We propose here an alternative method to determine the slope $K_L$. The downwelling scattered SW radiance $L_s^{\downarrow}$ has been calculating using *STREAMER* with the optical depth $\tau_{\mathrm{NICE}}$ determined in Sect. 4.1. This method is applied for each measured value of $B$, giving a total of 20 points. Results are shown in Fig. 7.

To ensure that the cloud cover is homogeneous in the area observed by the lidar and the radiometers (with different fields of
view), a linear regression between $B$ and $L_s^{\downarrow}$ has been computed only for opaque clouds (Fig. 7). The Pearson correlation coefficient is found to be $0.97$. Since the intercept $b$ is not zero ($b = 3143.53$), the relation between the measured solar background $B$ and the downwelling scattered SW radiance $L_s^{\downarrow}$ can be written as :

$$(B - b) = K_L L_s^{\downarrow} \tag{11}$$

with $K_L = 118.54 \ \mathrm{W}^{-1} \ \mathrm{m}^2 \ \mathrm{sr}$ retrieved from the fitting procedure. The vast majority of the cases with opaque clouds can be
correctly represented by this linear relationship. Since semi-transparent and opaque clouds represent about $83\%$ of the cloudy situations met during the drift (Sect 4.1), and because we did not observe any dependency on the solar zenith angle $\theta$ to build the





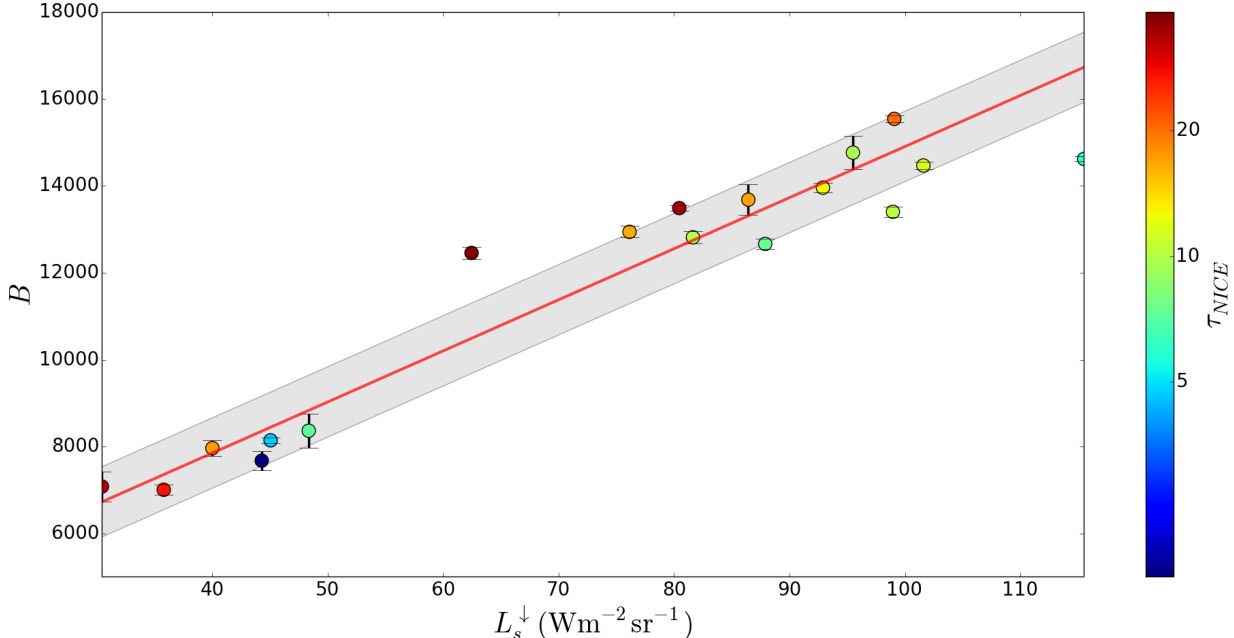

**Figure 7.** Solar background $B$ measured by IAOOS lidar as a function of the downwelling scattered SW radiance $L_s^{\downarrow}$ estimated with *STREAMER*. The color scale represents the cloud optical depth $\tau_{\mathrm{NICE}}$ obtained from the N-ICE observations. The error bars show the standard deviation on the solar background values. The solid red line shows the linear fit of the data points when $\tau_{\mathrm{NICE}} > 9$. The gray shaded area shows the dispersion from that regression line using the standard deviation on $B$ of 810 (Appendix A).

linear regression, the constant $K_L$ is applied to the whole dataset sampled aboard the IAOOS buoys. The spectral dependency of the solar scattered radiance calculated in the range $200 - 3600\,\mathrm{nm}$ is implicitly embedded into this slope $K_L$.

### 4.3 Determination of the cloud optical depth from the IAOOS lidar : $\tau_{\mathrm{IAOOS}}$

At any instant, the solar zenith angle $\theta$ being known, $L_s^{\downarrow}$ can be obtained from the solar background $B$ detected by the IAOOS lidar from Eq.11 (Sect. 4.2). Figure 4 highlights that for a given value of $L_s^{\downarrow}$ and a known value of $\theta$, there are two possible solutions for the cloud optical depth $\tau$, delimited by the threshold value $\tau_{\max}(L_s^{\downarrow})$ (Sect. 3.1). Those two solutions correspond a transparent cloud ($\tau < \tau_{\max}$) and a semi-transparent to opaque cloud ($\tau > \tau_{\max}$). In a similar approach applied on warm mid-latitude continental low-level clouds in ARM Oklahoma, Chiu et al. (2014) proposed a threshold value of $\tau_{\max} = 5$. This

threshold value is not appropriate here on cold low-level mixed-phase clouds in the Arctic Ocean, as $L_s^{\downarrow}$ peaks at lower values of $\tau_{\max}$ between $0.4$ and $4$ (Fig. 4). To remove this ambiguity, a first guess of the cloud optical depth ($\tau_0$) is needed. This latter is inferred from the maximum backscatter coefficient retrieved from the lidar range-corrected vertical profile, as in Di Biagio et al. (2020). In each cloud layer detected between the base $z_b$ and the top $z_t$ in the visible (similar to the base and top infrared





**Table 1.** Frequencies of the cloud optical depth $\tau$ derived from the N-ICE data and co-localized in time with the IAOOS data. Three groups are defined by their influence on the direct SW irradiance: transparent clouds ($\tau \leq 0.4$), semi-transparent clouds ($\tau \in ]0.4 - 4[$) and opaque clouds ($\tau \geq 4$). (Appendix B1)

| $\tau$ | $< 0.4$ | $]0.4 - 4[$ | $> 4$ |
|---|---|---|---|
| N-ICE | 0.17 | 0.24 | 0.59 |
| N-ICE co-localized | 0 | 0.11 | 0.89 |
| IAOOS co-localized | 0 | 0.07 | 0.93 |

in Sect. 3.3), the first guess of the cloud optical depth $\tau_0$ is simply obtained as :

$$\tau_0 = S_c \beta_{\max} \frac{z_t - z_b}{2} \tag{12}$$

where $S_c$ is the lidar ratio in the cloud layer and $\beta_{\max}$ is the maximal value of the backscatter coefficient $\beta$ within a vertical profile. The limit of that approach is that it permits the determination of $\tau$ only when $z_t$ can be identified, which generally occurs as long as the cloud optical depth does not reach $\tau \approx 3$ (Maillard et al., 2021). This approach is therefore not suitable to provide an accurate estimate of $\tau$ for opaque clouds, but it offers the opportunity to provide a rough information on the optical

properties of the cloud and to determine whether $\tau_{\mathrm{IAOOS}}$ is larger or lower than the threshold value $\tau_{\max}(\theta)$ seen in Fig. 4. Hence, this first guess helps to decide which of the two possible solutions is actually observed, permitting a determination of $\tau_{\mathrm{IAOOS}}$ from the knowledge of $L_s^\downarrow$.

Figure 6 presents the time series of $\tau_{\mathrm{IAOOS}}$ and $z_c$ in spring 2015. Frequent low-level clouds (below $500$ m altitude) with a strong optical depth between $10$ and $50$ are observed by the IAOOS lidar in spring. The average value of $\tau_{\mathrm{IAOOS}}$ is $17 \pm 8$,

which is slightly higher than that found using the radiometers as part of the N-ICE field experiment. Clouds are detected in the proximate vicinity of the surface, at altitudes between $200$ and $600$ m, in agreement with values reported by McFarquhar et al. (2007). A single cloud with an effective altitude $z_c > 2$ km is observed the 16 May and a couple of opaque clouds can even reach the surface level at the beginning of June. The latter are likely fog episodes observed by the lidar.

### 4.4 Comparison of the values of the cloud optical depth from the two methods

The frequencies of $\tau_{\mathrm{NICE}}$ and $\tau_{\mathrm{IAOOS}}$ determined from N-ICE and IAOOS measurements co-localized in time, corresponding to three classes are presented in Table 1. The three categories (transparent, semi-transparent and opaque clouds) are defined from the values of the SW irradiance at $\theta = 74\,°$ (shown on Fig. 4). Transparent clouds with $\tau < 0.4$ are characterized by a stronger direct SW irradiance then scattered SW irradiance. When $\tau$ is comprised between $0.4$ and $4$, clouds are called semi-transparent. In this regime, the direct SW beam is reduced and as a consequence the total SW downwelling irradiance is mostly controlled by

its scattered component, which is close to its maximum. When $\tau$ is above $4$ (opaque clouds), the direct SW beam is negligible with a value less then $0.5$ W m$^{-2}$ and $F_{\mathrm{SW}}^\downarrow$ is much less sensitive to changes in the value of $\tau$ (Appendix B1). Transparent clouds were not reported in spring 2015, using the observations from the lidar systems as part of the IAOOS campaigns or





the observations from the radiometers during N-ICE at the same time as the IAOOS measurements. In contrast, $17\%$ of the cases are classified as transparent clouds using all N-ICE measurements (Sect. 4.1). The whole set N-ICE radiometric data in

spring (Sect. 4.1) presents a distribution shifted towards semi-transparent and opaque clouds : $59\%$ of the clouds are classified as opaque clouds and $24\%$ as semi-transparent clouds. The frequency of opaque clouds obtained from N-ICE measurements is similar to the value ($60\%$) estimated by Zuidema et al. (2005) over a period of 10 days in May during SHEBA. Conversely, table 1 shows similar frequencies from the observed clouds for IAOOS and N-ICE taken at the same time, with opaque clouds $\sim 90\%$ of the time and semi-transparent clouds $\sim 10\%$ of the time. Both datasets lead to very similar results with a discrepancy

of only $4\%$ (Table 1). Using a direct inversion of the lidar backscatter signals obtained from the IAOOS buoys during all drifts, Maillard et al. (2021) estimated much lower cloud optical depths ($< 3$). It however does not reveal a disagreement with our findings as the values reported by Maillard et al. (2021) only focus on the first low-level cloud layer detected by the lidar, ignoring opaque clouds (that could completely attenuate lidar signals) or multilayered clouds.

Figure 6 confirms the agreement between the mean values of the optical depth, but with strong individual discrepancies

between $\tau_{IAOOS}$ and $\tau_{NICE}$ taken at the exact same time (mean bias of $0.98$). The average cloud optical depth in spring 2015 is estimated to be $16 \pm 10$ when N-ICE data are used and $17 \pm 8$ from the IAOOS lidar observations. The largest differences between the optical depths estimated from the two datasets lead to misclassified cloud types (semi-transparent versus opaque clouds).

The discrepancies between $\tau_{NICE}$ and $\tau_{IAOOS}$ may be ascribed to two main reasons : a difference in the field of view of the two

instruments and the role played by frost deposition on the lidar window. Chiu et al. (2014) also reported a correct representation of the optical depth, but they underlined discrepancies when compared to a radiometer that they attributed to the low field of view of a lidar. The radiometers (pyranometer and pyrgeometer instruments) measure the downwelling irradiances over a full hemisphere ($2\pi$ sr) whereas the downwelling scattered radiance $L_s^{\downarrow}$ used to calculate $\tau_{IAOOS}$ is derived from the backgound noise observed by the lidar with a narrow field of view of $0.67$ mrad ($\sim 3.5 \times 10^{-7}$ sr) (Mariage et al., 2017). The instruments

are close to each other (distance lower than $< 300$ m). Hence, heterogeneous structures of a cloud may lead to differences in the observed irradiances by the various instruments. Because of its very narrow field of view and since clouds are often detected in the lowest layers of the troposphere, the lidar can only detect an homogeneous portion of a single cloud. Radiometers used during the N-ICE campaign however sample a very broad area of the sky that may encompass clear sky zones and multiple inhomogeneous clouds. Notwithstanding, visual inspection during the campaign mostly revealed homogeneous cloud

conditions observed during the spring period (Walden et al., 2017), as as a consequence this hypothesis is likely insufficient to explain all differences between $\tau_{NICE}$ and $\tau_{IAOOS}$. The observed discrepancy may also be due to the formation of frost on the window of the receptor system. Signals obtained whit a frost index lower than $0.7$ have been discarded, but frost may cover a smaller fraction of the receptor window and strongly attenuate the intensity of the signal, underestimating the solar background. In turn, $L_s^{\downarrow}$ derived from Eq. 11 may also be underestimated. According to Fig. 4, this underestimation of the

downwelling radiances caused by the deposition of frost on the lidar window would be compensated (at a constant solar zenith angle) leading to an overestimation of the cloud optical depth.



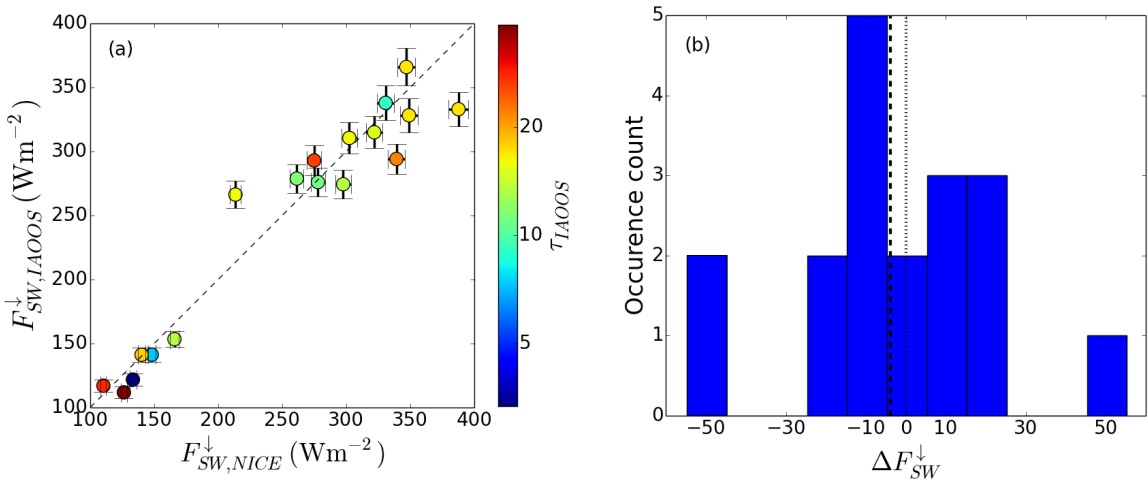

**Figure 8.** (a) Comparison of the downwelling SW irradiances derived from the IAOOS lidar measurements $F^{\downarrow}_{\text{SW, IAOOS}}$ and measured by the N-ICE radiometers $F^{\downarrow}_{\text{SW, NICE}}$ during the spring period. The color scale represents $\tau_{\text{NICE}}$. Vertical and horizontal error bars represent the uncertainties on $F^{\downarrow}_{\text{SW, IAOOS}}$ and $F^{\downarrow}_{\text{SW, NICE}}$ respectively. The 1:1 reference line is plotted as a dotted line. (b) Histogram of the absolute differences between $F^{\downarrow}_{\text{SW, IAOOS}}$ and $F^{\downarrow}_{\text{SW, NICE}}$. The dashed line represents the mean difference.

## 4.5 Estimation of $F^{\downarrow}_{\text{SW}}$ and $F^{\downarrow}_{\text{LW}}$ irradiances

The determination of the cloud optical depth from the measured solar background of the lidar (Sect. 4.3) enables to calculate $F^{\downarrow}_{\text{SW,s}}(\tau_{\text{IAOOS}}, \theta)$ using *STREAMER* and finally $F^{\downarrow}_{\text{SW}}(\tau_{\text{IAOOS}}, \theta)$ from Eq. 1. This downwelling SW irradiance $F^{\downarrow}_{\text{SW, IAOOS}}$

335 is further compared to that directly measured by the radiometers as part of the N-ICE 2015 campaign ($F^{\downarrow}_{\text{SW, NICE}}$). The estimated uncertainty on $F^{\downarrow}_{\text{SW, IAOOS}}$ is $\pm 4\%$ (Appendix A). The comparison and differences between those two downwelling SW irradiances are reported in Fig. 8a and b.

 Figure 8a generally highlights a good agreement between the estimated $F^{\downarrow}_{\text{SW, IAOOS}}$ and the measured $F^{\downarrow}_{\text{SW, NICE}}$ irradiances, with a Pearson correlation coefficient of 0.96. The methodology introduced here to estimate the SW irradiances from the

340 measured solar background of the IAOOS lidar therefore reproduces the SW irradiances directly measured by radiometers with an uncertainty close to $9\,\text{W}\,\text{m}^{-2}$ (Appendix A). The absolute differences between $F^{\downarrow}_{\text{SW, IAOOS}}$ and $F^{\downarrow}_{\text{SW, NICE}}$ are represented in Fig. 8b. These results highlight a small mean bias of $-3.7\,\text{W}\,\text{m}^{-2}$, even though strong discrepancies are observed for a few outlier showing an absolute differences of $50\,\text{W}\,\text{m}^{-2}$.



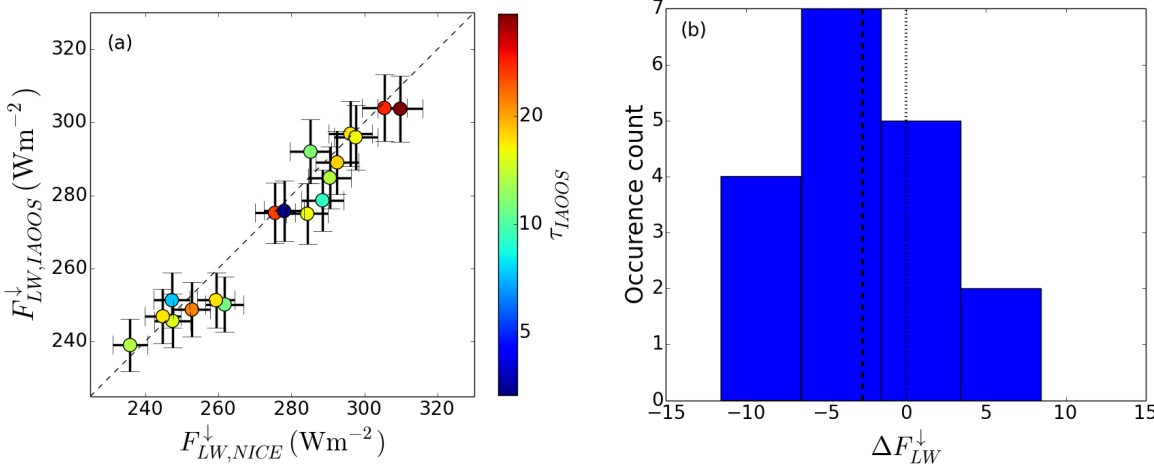

**Figure 9.** (a) Comparison of the downwelling LW irradiances derived from the IAOOS lidar measurements $F^\downarrow_{\text{LW, IAOOS}}$ and measured by the N-ICE radiometers $F^\downarrow_{\text{LW, NICE}}$ during the spring period. Vertical and horizontal error bars represent the uncertainties on $F^\downarrow_{\text{LW, IAOOS}}$ and $F^\downarrow_{\text{LW, NICE}}$ respectively. The 1:1 reference line is plotted as a dotted line. (b) Histogram of the absolute differences between $F^\downarrow_{\text{LW, IAOOS}}$ and $F^\downarrow_{\text{LW, NICE}}$. The dashed line represents the mean difference.

The downwelling LW irradiance is estimated using the methodology described in Sect. 3.3. In particular the IAOOS lidar
measurements help to determine the cloud mask $c$, the effective altitude of the cloud $z_c$ (Sect. 3.3) and the optical depth of
the cloud $\tau_{\text{IAOOS}}$ (Sect. 4.3). That latter is used to determine $\epsilon_c$ through Eq. 7 and finally the downwelling LW irradiance
$F^\downarrow_{\text{LW,IAOOS}}$ thanks to Eq. 5. This radiative flux is then compared to the downwelling LW irradiance $F^\downarrow_{\text{LW,NICE}}$ directly measured
by the pyrgeometer during the N-ICE campaign. The results are shown in Fig. 9a. The estimated uncertainty on $F^\downarrow_{\text{LW, IAOOS}}$ is 9
$\text{W m}^{-2}$ (Appendix A).

A fairly good correlation is obtained between the estimated $F^\downarrow_{\text{LW,IAOOS}}$ and the measured $F^\downarrow_{\text{LW,NICE}}$ LW irradiances, with a
Pearson correlation coefficient of $0.98$. The differences, shown in Fig. 9b, are always lower than $12 \text{ W m}^{-2}$, with a corrigible
residual bias of $3 \text{ W m}^{-2}$. The approach relying on IAOOS lidar observations to derive LW downwelling irradiances, provides
a good estimation of the fluxes within an uncertainty range of $9 \text{ W m}^{-2}$.

## 4.6   Discrepancies of $F^\downarrow_{\text{SW}}$ and $F^\downarrow_{\text{LW}}$ irradiances

The absolute differences between the estimated and measured SW irradiances $\Delta F^\downarrow_{\text{SW}}$ as a function of the absolute difference
between $\tau_{\text{IAOOS}}$ and $\tau_{\text{NICE}}$ are shown in Fig. 10a. $\Delta \tau$ decreases monotonically with $\Delta F^\downarrow_{\text{SW}}$ : when the optical depth is over-





estimated (resp. underestimated), the downwelling SW irradiance is underestimated (resp. overestimated). A decrease in the downwelling scattered SW irradiance associated to a larger optical depth indeed leads to an underestimated total downwelling SW irradiance. In our study, as cloud optical depths are always large ($\tau_{\mathrm{IAOOS}} > 3$), $F_{\mathrm{SW,s}}^{\downarrow}$ therefore depends linearly on $L_s^{\downarrow}$

(Fig. 5). The values of $L_s^{\downarrow}$ used in Fig. 7 to obtain Eq. 11 were retrieved from *STREAMER* assuming that clouds have an optical depth equal to $\tau_{\mathrm{NICE}}$ (Sect. 4.2). The absolute difference $\Delta\tau$ is thus directly linked to the differences between the regression line and the $L_s^{\downarrow}$ values obtained from *STREAMER* at $\tau_{\mathrm{NICE}}$ on Fig. 7, confirming the decreasing trend observed in Fig. 10a. Actually, according to Fig. B1, $F_{\mathrm{SW}}^{\downarrow}$ varies linearly with the logarithm of the cloud optical depth ; hence the relationship between the relative error on $\tau$ and the absolute error on $F_{\mathrm{SW}}^{\downarrow}$ is quasi-linear. Frost deposition on the lidar window and the difference in the

field of view of the two instruments are the two main reasons explaining the discrepancies between $\tau_{\mathrm{IAOOS}}$ and $\tau_{\mathrm{NICE}}$ (sect. 4.4), which in turn are responsible of the differences between $F_{\mathrm{SW,\ IAOOS}}^{\downarrow}$ and $F_{\mathrm{SW,\ NICE}}^{\downarrow}$. The cloud properties (type and size of particle, geometrical height, etc) do not participate in the observed discrepancies since both $\tau_{\mathrm{IAOOS}}$ and $\tau_{\mathrm{NICE}}$ are estimated from the same modeled mixed-phase cloud. Some of the outliers, represented on Fig. 8b and Fig. 10a are characterized by strong differences between $F_{\mathrm{SW,\ IAOOS}}^{\downarrow}$ and $F_{\mathrm{SW,\ NICE}}^{\downarrow}$ that can reach $\pm 50\ \mathrm{W\,m^{-2}}$. They cannot be explained by frost deposition on the

window, as solar background signals significantly attenuated by frost have been removed (frost index lower than $0.7$). On the contrary, the fact that the two instruments (lidar and radiometer) with different fields of view see distinct atmospheric scenes, as suggested in Fig. 7, leads to differences between $\tau_{\mathrm{IAOOS}}$ and $\tau_{\mathrm{NICE}}$ (Fig. 10a). As a result, an underestimation (resp. overestimation) of $F_{\mathrm{SW,\ IAOOS}}^{\downarrow}$ is observed as soon as $\Delta\tau > 0$ ($\Delta\tau < 0$). Outliers are represented by strong absolute differences of $\Delta\tau = 11$ and $\Delta\tau = -22$. When those outliers are excluded from the analysis, our methodology based on IAOOS lidar mea-

surements provides good estimations of the SW irradiances with a mean bias of $-1.5\ \mathrm{W\,m^{-2}}$. However, due to the difference in the observed atmospheric scenes, the RMSE $\approx 24\ \mathrm{W\,m^{-2}}$ is larger than the uncertainty of the determination of $F_{\mathrm{SW,\ IAOOS}}^{\downarrow}$ ($9\ \mathrm{W\,m^{-2}}$, Appendix A). A longer lidar sequence acquisition time could possibly smooth out cloud heterogeneity and improve the determination of $F_{\mathrm{SW}}^{\downarrow}$.

Maximum discrepancies between the estimated and the measured LW irradiances can reach $12\ \mathrm{W\,m^{-2}}$ (Fig. 9b). The

fact that all LW irradiance values reported on Fig. 9a are larger than $230\ \mathrm{W\,m^{-2}}$ suggests that the lidar system detected clouds ($c = 1$) for the entirety of the spring period (Table 1). Equation 5 shows that $T_s$ has therefore a very limited role on the determination of $F_{\mathrm{LW,IAOOS}}^{\downarrow}$. Hence the differences between the estimated and measured fluxes can be explained by two possible factors : either by errors in the optical properties of the clouds detected by the lidar ($\tau_{\mathrm{IAOOS}}$) or by an incorrect estimation of the cloud temperature $T_c$. The cloud optical depth drives the emissivity of the cloud $\epsilon_c$. Since, the lidar detected almost exclusively

opaque clouds (Table 1), clouds are considered as black bodies on $93\%$ of the cases and $\epsilon_c = 1$. The error that would be caused by an incorrect determination of the cloud optical depth is very weak: the distribution of $\tau_{\mathrm{IAOOS}}$ in Table 1 reveals that $7\%$ of the clouds have an optical depth lower than $4$. Even when $\tau_{\mathrm{IAOOS}} \in [0.4, 4]$, $\epsilon$ remains close to $1$ and the LW irradiance is not terribly modified. Figure 10b confirms that the discrepancy on the optical depth $\Delta\tau$ does not significantly modify the downwelling LW irradiance. Only small differences in the temperature at cloud height between ERA5 and the N-ICE radiosoundings at cloud

height were noticed (Fig. 3). Those discrepancies are characterized by a mean bias of $-0.3\ \mathrm{K}$ and a RMSE of $1.6\ \mathrm{K}$. The only significant cause of the discrepancies between $F_{\mathrm{LW,IAOOS}}^{\downarrow}$ and $F_{\mathrm{LW,NICE}}^{\downarrow}$ is thus the effective altitude of the cloud $z_c$ that





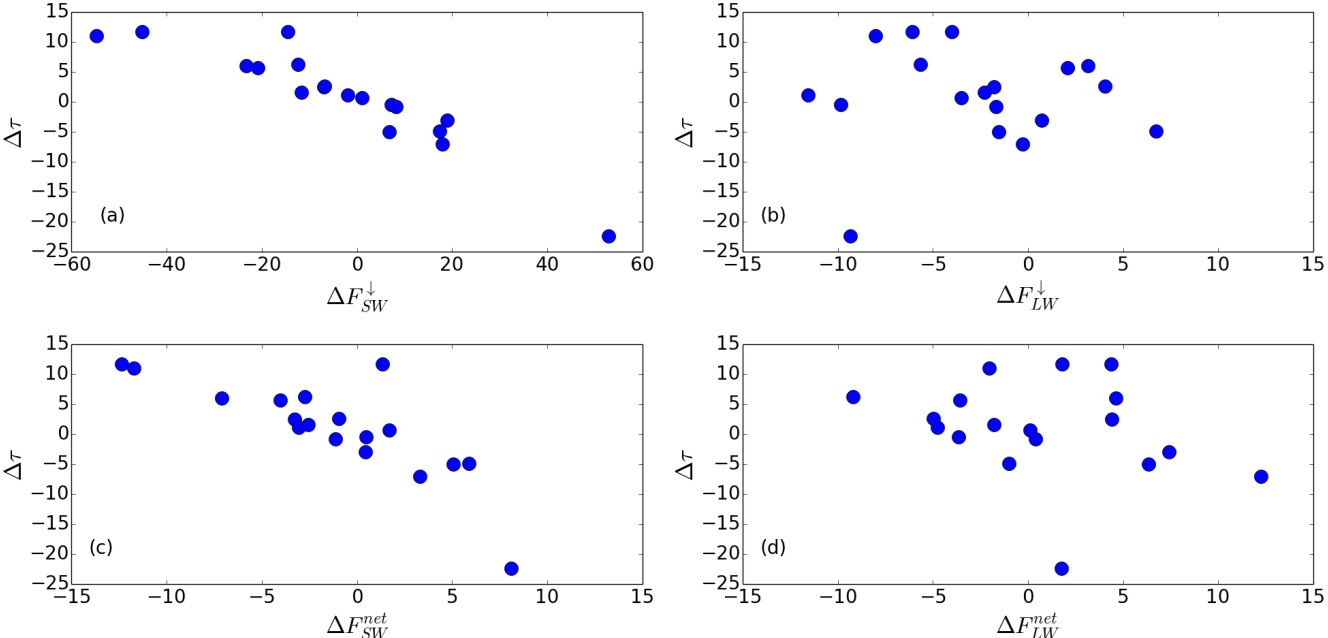

**Figure 10.** Absolute differences between the estimated and the N-ICE measured downwelling (a,b) and net (c,d) SW (a,c) and LW (b,d) irradiances as a function of the absolute difference on the cloud optical depth $\Delta\tau$.

drives the effective temperature of the cloud $T_c$. During the spring period, we estimated an average value of $T_c \simeq 261 \pm 5$ K. The effective temperature of the cloud $T_c$ is however very sensitive to the altitude of the cloud $z_c$, which may present spatial heterogeneities. A difference of $\Delta z_c = 200$ m leads to $\Delta T_c = 2°$C and, in turn, to a variation of $\Delta F_{\mathrm{LW}}^{\downarrow} = 4\epsilon_c \sigma T_c^3 \Delta T_c \approx 8.7$

W m$^{-2}$ on the downwelling LW irradiance.

## 4.7 Estimation of net SW $F_{\mathrm{SW}}^{\mathrm{net}}$ and LW $F_{\mathrm{LW}}^{\mathrm{net}}$ irradiances

The net SW $F_{\mathrm{SW}}^{\mathrm{net}}$ and net LW $F_{\mathrm{LW}}^{\mathrm{net}}$ irradiances are calculated as the difference between the downwelling and the upwelling irradiances. Using Eq 3 and Eq 6, $F_{\mathrm{SW,IAOOS}}^{\uparrow}$ and $F_{\mathrm{LW,IAOOS}}^{\uparrow}$ are both estimated with uncertainties of 9 W m$^{-2}$ (Appendix A).

The absolute difference $\Delta F_{\mathrm{SW}}^{\mathrm{net}}$ between the estimated net SW irradiance and its counterpart observed during the N-ICE

experiment are represented in Fig. 10c. $\Delta F_{\mathrm{SW}}^{\mathrm{net}}$ presents a mean bias of $-1$ W m$^{-2}$ and a RMSE of 5 W m$^{-2}$, within the uncertainty range of the method developed here to estimate the SW irradiances. The relationship between $\Delta F_{\mathrm{SW}}^{\mathrm{net}}$ and $\Delta\tau$ follows the same tendency than that previously discussed for $\Delta F_{\mathrm{SW}}^{\downarrow}$ in Sect. 4.5. $\Delta\tau$ decreases monotonically with $\Delta F_{\mathrm{SW}}^{\mathrm{net}}$. An underestimation of $F_{\mathrm{SW,IAOOS}}^{\mathrm{net}}$ is observed when $\Delta\tau > 0$ and, in contrast, an overestimation is seen when $\Delta\tau < 0$. The discrepancy between the estimated and measured net irradiances is systematically lower than $\Delta F_{\mathrm{SW}}^{\downarrow}$ discussed in Fig. 10a,

even for the outliers. It suggests that the differences between the estimated and measured upwelling components $\Delta F_{\mathrm{SW}}^{\uparrow}$ partly compensate the differences on the downwelling irradiances $\Delta F_{\mathrm{SW}}^{\downarrow}$. An overestimation (resp. underestimation) of $F_{\mathrm{SW, IAOOS}}^{\downarrow}$



over $F_{\mathrm{SW,\,NICE}}^{\downarrow}$ leads to a quasi-proportional overestimation (resp. underestimation) of $F_{\mathrm{SW,IAOOS}}^{\uparrow}$ over $F_{\mathrm{SW,NICE}}^{\uparrow}$. The surface albedo is supposed to be very similar on the locations of the two instruments and, consequently, the difference on $\Delta F_{\mathrm{SW}}^{\downarrow}$ is partly canceled out by a proportional difference on $\Delta F_{\mathrm{SW}}^{\uparrow}$. This approach enables to estimate the net SW irradiance from solar

background measurement by a lidar with an uncertainty lower than $13\ \mathrm{W\,m^{-2}}$, in good agreement with the net SW irradiance observations from the N-ICE experiment, with a Pearson correlation coefficient of $0.95$.

Conversely, the absolute differences on the net LW irradiances $\Delta F_{\mathrm{LW}}^{\mathrm{net}}$ (Fig. 10d) are higher than the absolute differences on the downwelling irradiances $\Delta F_{\mathrm{LW}}^{\downarrow}$ discussed in Sect. 4.5. The mean bias on $\Delta F_{\mathrm{LW}}^{\mathrm{net}}$ is $2\ \mathrm{W\,m^{-2}}$ and the associated RMSE is $5\ \mathrm{W\,m^{-2}}$. The absolute differences are mostly positive and may reach $11\ \mathrm{W\,m^{-2}}$, indicating that $F_{\mathrm{LW,IAOOS}}^{\uparrow} \leq F_{\mathrm{LW,NICE}}^{\uparrow}$,

which partly compensate $\Delta F_{\mathrm{LW}}^{\downarrow}$. The discrepancies on the cloud optical depth $\Delta\tau$ do not affect the net LW irradiance, since all clouds are opaque or semi-transparent with $\tau_{\mathrm{IAOOS}}$ above 3, acting as black bodies. In addition, the 2-m air temperature does not either play any role on $F_{\mathrm{LW,IAOOS}}^{\mathrm{net}}$ given that episodes of clear sky were never observed by the lidar (Eq. 5). The surface emissivity is not sufficient to explain the underprediction of $F_{\mathrm{LW,IAOOS}}^{\uparrow}$. The only possible explanation is due to errors on the skin temperature. The skin temperature taken from the ERA5 reanalysis may be colder than the actual skin temperature. A

difference of $\Delta T_s = 2^\circ\mathrm{C}$ leads to a difference on the upwelling LW irradiance of $9\ \mathrm{W\,m^{-2}}$. The uncertainty on $F_{\mathrm{LW}}^{\mathrm{net}}$ caused by the use of the ERA5 reanalysis does not exceed the uncertainly range of $13\ \mathrm{W\,m^{-2}}$ (Appendix A). This approach provides a reasonable estimate of the net LW irradiance from temperature and lidar measurements with a Pearson correlation coefficient of $0.7$.

Both the net SW and LW irradiances are well reproduced when using the IAOOS lidar measurements. The total net irradiance

$(F_{\mathrm{SW}}^{\mathrm{net}} + F_{\mathrm{LW}}^{\mathrm{net}})$ can be similarly determined from lidar measurements. A fairly good agreement (not shown) with the total net irradiance (SW+LW) measured during the N-ICE campaign is found, with a mean bias of $-0.5\ \mathrm{W\,m^{-2}}$ and a RMSE of $8$ $\mathrm{W\,m^{-2}}$.

## 5  Conclusions

To complete the need for more data on the Arctic atmosphere beyond $80^\circ\mathrm{N}$, multiple drifting buoys were deployed in the

Arctic Ocean between 2014 and 2019 as part of the IAOOS project. But those buoys were not equipped with instruments capable of measuring the different components of the surface energy budget. In this study we investigate the possibility to extend the IAOOS dataset to include estimated radiative fluxes on buoys. We present an alternative method to estimate the downwelling shortwave (SW) and longwave (LW) irradiances from the knowledge of the cloud optical depth $\tau$ and effective cloud height $z_c$ derived from zenith-pointing backscatter lidar measurements and using radiative transfer model outputs. We

apply this methodology to observations carried out in spring (April-June) 2015 in the vicinity of the N-ICE field expedition, north of Svalbard in order to define its applicability, limits, biases and uncertainties. Cloud optical depths $\tau_{\mathrm{NICE}}$ and $\tau_{\mathrm{IAOOS}}$ are estimated from the downwelling SW irradiance measurements during N-ICE and from the solar background noise measured by the IAOOS lidar, with an uncertainty of $0.61$ and $0.85$, respectively. In springtime, opaque clouds ($\tau > 4$) were detected $59\%$ of the time. This frequency is only $24\%$ for semi-transparent clouds ($\tau \in [0.4 - 4]$) and $17\%$ for transparent clouds or





clear sky conditions ($\tau < 0.4$). Taking into account only instants when lidar measurements were performed, this distribution is even more shifted toward opaque clouds: $\tau_{\mathrm{NICE}}$ and $\tau_{\mathrm{IAOOS}}$ present very similar frequencies of $89\%$ and $93\%$ for opaque clouds and $11\%$ and $7\%$ for semi-transparent clouds. The small differences between $\tau_{\mathrm{NICE}}$ and $\tau_{\mathrm{IAOOS}}$ can be ascribed either to frost formed on the window of the receptor or to differences in the atmospheric scenes seen by the two instruments (lidar and radiometer) having different fields of view.

The downwelling SW irradiance is assessed from the solar background measured by the IAOOS lidar with an uncertainty of 9 $\mathrm{W\,m^{-2}}$. Discrepancies noted between the two instruments are linked to their different fields of view so that distinct atmospheric scenes are observed or to frost forming on the window of the lidar. An overestimation (resp. underestimation) of $\tau$ will lead to an underprediction (resp. overprediction) of the downwelling SW irradiance. An decrease in the downwelling scattered SW irradiance associated to a larger optical depth leads to an underestimated total downwelling SW irradiance and a RMSE of

24 $\mathrm{W\,m^{-2}}$. The downwelling LW irradiance is estimated from a simple model with three fluxes using the skin temperature, the 2-m air temperature and the temperature of emission of the cloud obtained from ERA5 reanalyses, with an uncertainty of 9 $\mathrm{W\,m^{-2}}$ and a RMSE of 5 $\mathrm{W\,m^{-2}}$. Because all skies are covered by either opaque or semi-transparent clouds ($\tau_{\mathrm{IAOOS}} > 3$), only the emission temperature at cloud height really controls the downwelling LW irradiance. Both the downwelling SW and LW irradiances, estimated from the IAOOS lidar, are in agreement with the measured values of the N-ICE field campaign in

spring 2015, with a small mean bias of $-1.5\ \mathrm{W\,m^{-2}}$.

   The net SW and LW irradiances are finally estimated with an uncertainty lower than 13 $\mathrm{W\,m^{-2}}$. The net SW irradiance assessed from the lidar measurements is reproduced with a lower bias ($-1\ \mathrm{W\,m^{-2}}$) and lower RMSE (5 $\mathrm{W\,m^{-2}}$) than its downwelling component (mean bias of $-1.5\ \mathrm{W\,m^{-2}}$ and RMSE of 24 $\mathrm{W\,m^{-2}}$). Indeed discrepancies on the downwelling SW irradiance are partly compensated by proportional differences on its upwelling counterpart (due to the surface albedo). The

upwelling LW irradiance is assessed from the skin temperature of ERA5, which may be colder than the actual skin temperature. The resulting net LW irradiance is partly overestimated, with a mean bias of 2 $\mathrm{W\,m^{-2}}$ and a RMSE of 5 $\mathrm{W\,m^{-2}}$. Measuring the skin temperature on the IAOOS buoy (not available in spring 2015) could improve the upward LW irradiance, as well as the net LW irradiance. The net SW and LW irradiances retrieved from the IAOOS lidar observations are in agreement with the measured values during the N-ICE field campaign in spring 2015.

This promising approach to estimate the downwelling, upwelling and net SW and LW irradiances from lidar measurements on drifting buoys together with radiative model calculations will be further applied to 6 years of measurements in the High Arctic region. This will provide the key components of the radiative budget on ice surfaces. Efforts are nevertheless needed to better correct the solar background measured by the lidar when frost is deposited on the receptor window in order to increase the number of observations.

*Code and data availability.* The N-ICE2015 observational data sets are available from the Norwegian Polar Data Center (https://data.npolar.no/dataset/). The IAOOS atmospheric data are available through the AERIS Data Portal at https://www.aerisdata.fr/catalogue/ (Ravetta and Pascal, 2018). The *STREAMER* radiative transfer model can be downloaded at https://stratus.ssec.wisc.edu/streamer/streamer.html.



## Appendix A: Uncertainties on the retrievals of the cloud optical depth and radiative fluxes

### A1  Uncertainties on $\tau$

The cloud optical depth is estimated by minimizing the absolute differences between the observed values of downwelling irradiances (Sect. 4.1) or scattered radiances (Sect. 4.3) and their simulated counterparts by *STREAMER*. $\tau_{\text{NICE}}$ is obtained from the observed of $F_{\text{SW}}^{\downarrow}$ measured by the radiometer of the N-ICE field expedition during the spring period, whereas $\tau_{\text{IAOOS}}$ is determined from $L_s^{\downarrow}$ estimated from solar background noise $B$ (Sect. 4.2). As a result, there are two sources of uncertainties in the determination of $\tau$ : the uncertainty $\Delta\tau^m$ due to the measured values of $F_{\text{SW}}^{\downarrow}$ or $L_s^{\downarrow}$ and the uncertainty $\Delta\tau^s$ due to the

model results triggered by the hypotheses on the surface and cloud parameters formulated in *STREAMER* (Sect. 3.1).

Hudson et al. (2016) and Walden et al. (2017) estimated a measurement error on the downwelling SW irradiance of $\Delta F_{\text{SW}}^{\downarrow} = 5\ \text{W}\,\text{m}^{-2}$ on the radiometer observations. The uncertainty on the downwelling SW scattered radiance can be calculated as $\Delta L_s^{\downarrow} = \dfrac{\Delta B}{K_{L}}$ using Eq. 11, where $\Delta B = 810$ is the uncertainty on the solar background noise. We therefore estimated $\Delta L_s^{\downarrow} = 6.75\ \text{W}\,\text{m}^{-2}\,\text{sr}^{-1}$. According to Fig. 4, the resulting uncertainties on the optical depth were assessed to $\Delta\tau_{\text{NICE}}^m = 0.14$ and

$\Delta\tau_{\text{IAOOS}}^m = 0.35$.

To assess the influence of the surface and cloud parameters used in our reference simulation (Fig. 4), we performed sensitivity tests of the dependence of $F_{\text{SW}}^{\downarrow}$ and $L_s^{\downarrow}$ on the surface albedo, the cloud phase, and the size of the hydrometeors (cloud droplets and ice crystals). A total of 54 simulations were run : the solar zenith angle was set to a constant value of $74°$, with different values for the surface albedo (0.78, 0.82, 0.86), the type of hydrometeors inside the first layer (plate-like or column-like ice

crystals), the diameter of ice crystals ($21.3\ \mu\text{m}$, $25.2\ \mu\text{m}$ and $29.1\ \mu\text{m}$) and the diameter of water droplets ($5.1\ \mu\text{m}$, $6.9\ \mu\text{m}$ and $8.7\ \mu\text{m}$). In each sensitivity test, the minimization of the absolute differences between the observed and modeled values of the irradiances or radiances lead to an estimation of $\tau$. The Root Mean Squared Error (RMSE) of the optical depth between those 54 sensitivity tests and the reference simulation (presented in Fig. 4) were estimated to $\Delta\tau_{\text{NICE}}^s = 0.59$ and $\Delta\tau_{\text{IAOOS}}^s = 0.77$.

Finally the total absolute uncertainty on $\tau$ is estimated by the relation $\Delta\tau = \sqrt{(\Delta\tau^m)^2 + (\Delta\tau^s)^2}$. We find $\Delta\tau_{\text{NICE}} = 0.61$

and $\Delta\tau_{\text{IAOOS}} = 0.85$.

### A2  Uncertainties on $F_{\text{SW,IAOOS}}^{\downarrow}$ and $F_{\text{SW,IAOOS}}^{\uparrow}$

According to Eq. 1, the uncertainty on the downwelling SW irradiance is estimated as $\Delta F_{\text{SW}}^{\downarrow} = \sqrt{\left(\Delta F_{\text{SW,d}}^{\downarrow}\right)^2 + \left(\Delta F_{\text{SW,s}}^{\downarrow}\right)^2}$. This latter term $\Delta F_{\text{SW,s}}^{\downarrow}$ depends on the uncertainties on the cloud optical depth $\Delta\tau_{\text{IAOOS}}$ (Sect. A1) and the uncertainty of $F_{\text{SW,s}}^{\downarrow}$ from the *STREAMER* radiative transfer model. It is estimated to $\Delta F_{\text{SW,s}}^{\downarrow} = 8.1\ \text{W}\,\text{m}^{-2}$. Following Eq. 2, the relative

uncertainty on the direct component of the downwelling SW irradiance is $\Delta F_{\text{SW,d}}^{\downarrow} = F_{\text{SW,d}}^{\downarrow} \dfrac{\Delta\tau_{\text{IAOOS}}}{\cos\theta} = 0.7\ \text{W}\,\text{m}^{-2}$. Finally, using Eq. 3 and because the surface albedo $\alpha$ is fixed, the relative uncertainties on the upwelling and downwelling components of the SW irradiances derived from the lidar observations are similar : $\dfrac{\Delta F_{\text{SW,IAOOS}}^{\uparrow}}{F_{\text{SW,IAOOS}}^{\uparrow}} = \dfrac{\Delta F_{\text{SW,IAOOS}}^{\downarrow}}{F_{\text{SW,IAOOS}}^{\downarrow}} = 0.04$.





## A3 Uncertainties on $F_{\text{LW,IAOOS}}^{\downarrow}$ and $F_{\text{LW,IAOOS}}^{\uparrow}$

Applying the variance formula on Eq. 5 enables to compute the uncertainty on the downwelling LW irradiance $\Delta F_{\text{LW,IAOOS}}^{\downarrow}$. In

cloudy situations, representing all cases during the drifts of the IAOOS buoys during spring 2015, it can be written as :

$$\Delta F_{\text{LW,IAOOS}}^{\downarrow} = \sqrt{\left|\frac{\partial F_{\text{LW,IAOOS}}^{\downarrow}}{\partial \epsilon_c}\right|^2 \Delta \epsilon_c^2 + \left|\frac{\partial F_{\text{LW,IAOOS}}^{\downarrow}}{\partial T_c}\right|^2 \Delta T_c^2 + \left|\frac{\partial F_{\text{LW,IAOOS}}^{\downarrow}}{\partial T_s}\right|^2 \Delta T_s^2} \tag{A1}$$

with $\Delta T_c = \Delta T_s = 2.2°C$ and $\Delta \epsilon_c = \exp\left(-\frac{\beta_1 \tau_{\text{IAOOS}}^{\beta_2}}{\cos\theta}\right)\beta_1\beta_2\frac{\tau_{\text{IAOOS}}^{\beta_2-1}}{\cos\theta}\Delta\tau_{\text{IAOOS}} = 0.0013$ the uncertainties on temperature at

cloud height, surface temperature and cloud emissivity, obtained from Eq. 7, respectively. $\Delta T_c$ is a function of the uncertainties

on the effective altitude $\Delta z_c$ and of the vertical profile of temperature obtained from the ERA5 reanalyses. $\Delta T_s$ only depends

the uncertainty on the temperature from ERA5 reanalyses. The partial derivatives can be computed as :

$$\frac{\partial F_{\text{LW,IAOOS}}^{\downarrow}}{\partial \epsilon_c} = \sigma T_c^4 - \epsilon_s \sigma T_s^4 \tag{A2}$$

$$\frac{\partial F_{\text{LW,IAOOS}}^{\downarrow}}{\partial T_c} = 4\epsilon_c \sigma T_c^3 \tag{A3}$$

$$\frac{\partial F_{\text{LW,IAOOS}}^{\downarrow}}{\partial T_s} = 4\epsilon_c \sigma (1-\epsilon_c) T_s^3 \tag{A4}$$

$$\tag{A5}$$

The global uncertainty on the downwelling LW irradiance is therefore $\Delta F_{\text{LW,IAOOS}}^{\downarrow} = 9 \text{ W m}^{-2}$. Finally, using Eq. 6, the

uncertainty on the upwelling LW irradiance can be simply written as $\Delta F_{\text{LW,IAOOS}}^{\uparrow} = 4\epsilon_s \sigma T_s^3 \Delta T_s = 9 \text{ W m}^{-2}$.

## Appendix B: Definition of cloud types

Figure B1 shows the direct and scattered downwelling SW irradiance computed using the *STREAMER* radiative transfer model

at a solar zenith angle $\theta = 74°$ for different values of $\tau$. Three categories of clouds (transparent, semi-transparent and opaque

clouds) are defined according to the values of the downwelling SW irradiance. Transparent clouds with $\tau < 0.4$ are character-

ized by a larger direct SW irradiance than its scattered counterpart. When $\tau$ is comprised between $0.4$ and $4$, clouds are called

semi-transparent. In this regime, the direct SW beam is reduced and as a consequence the total SW downwelling irradiance is

mostly controlled by its scattered component, which is close to its maximum. When $\tau$ is above 4 (opaque clouds), the direct

SW beam is negligible with a value less then $0.5 \text{ W m}^{-2}$ and $F_{\text{SW}}^{\downarrow}$ is much less sensitive to changes in the value of $\tau$.

*Author contributions.* LL performed the data treatment and analysis and prepared the manuscript. JCR and JP provided supervision, guidance

and editing. JP led the IAOOS project and designed the lidar. VM constructed the lidar. VM, CDB, JM contributed the treatment of the lidar

data.





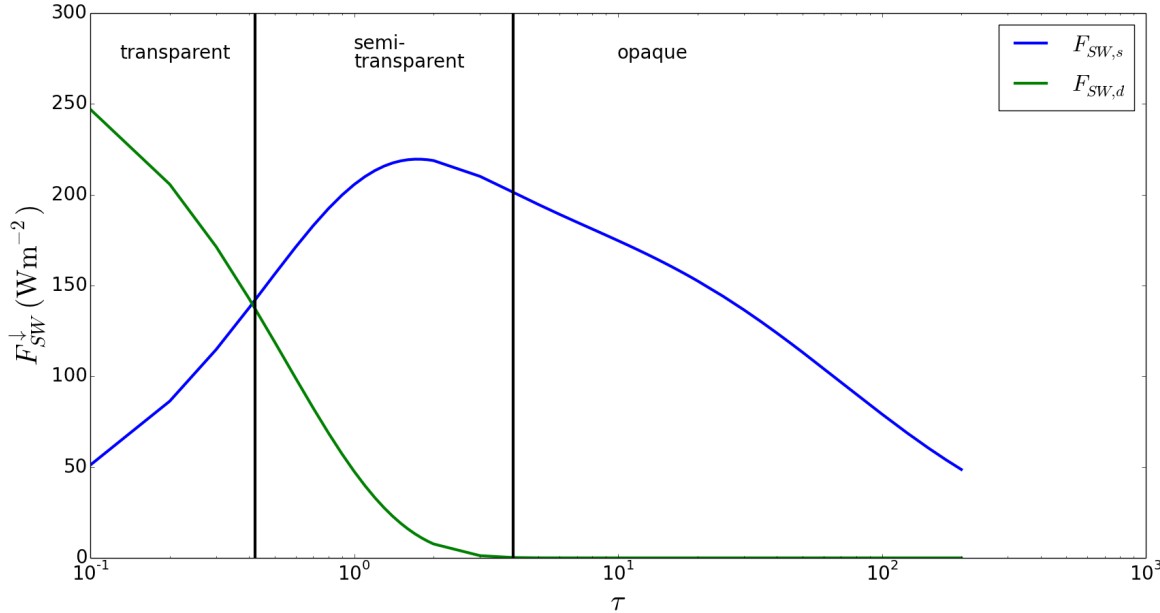

**Figure B1.** Direct (green) and scattered (blue) downwelling SW irradiance computed using the *STREAMER* radiative transfer model at a solar zenith angle $\theta = 74°$. Three categories of clouds (transparent, semi-transparent and opaque clouds) are defined according to the values of the downwelling SW irradiance.

*Competing interests.* The authors declare that they have no conflict of interest.

*Acknowledgements.* The authors acknowledge the Norwegian Polar Institute's Centre for Ice, Climate and Ecosystems (ICE) through the
N-ICE project for supporting the N-ICE2015 campaign. They also acknowledge the support of Equipex IAOOS (Ice Atmosphere Ocean Observing System, ANR-10-EQPX-32-01) for the development of the buoys in partnership with LOCEAN at Sorbonne University. The IAOOS campaign was from the ICE-ARC programme from the European Union Seventh Framework Programme grant number 603887. Computer analyses benefited from access to IDRIS HPC resources (GENCI allocations A007017141 and A009017141) and the IPSL mesoscale computing center (CICLAD: Calcul Intensif pour le CLimat, l'Atmosphère et la Dynamique). Our results based on ERA5 analyses contain
modified Copernicus Climate Change Service information 2020. Neither the European Commission nor ECMWF is responsible for any use that may be made of the Copernicus information or data it contains.



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
