# Peer review of "Radiative fluxes in the High Arctic region derived from ground-based lidar measurements onboard drifting buoys"

_Atmospheric Measurement Techniques, 2021_

## Referee Comment (RC2)

Review of amt-2021-326: "**Radiative fluxes in the High Arctic region derived from ground-based lidar measurements onboard drifting buoys**" by Lilian Loyer, Jean-Christophe Raut, Claudia Di Biagio, Julia Maillard, Vincent Mariage, and Jacques Pelon.

Ian Brooks

**Overview**

This paper documents an initial evaluation of whether the background noise from the lidars installed on IAOOS drifting buoys in the Arctic Ocean can be used to estimate at least some of the components of the surface radiation budget. The approach relies upon combination of the lidar noise measurement combined with information from a radiative transfer model, along with some input from ERA5 reanalysis fields. The method progresses through a number of steps, some of which depend upon assumptions about conditions. The impact of these assumptions is only explored for some of them, and the uncertainties introduced by the others remain unquantified.

While well written overall, the method described is complicated, and there are a few places where the discussion becomes confusing and would benefit from being clarified – this often results from a term being used before it is defined – these are noted below. There are also a number of grammatical slip-ups and typos, detailed below.

**Major comments**

While the approach documented appears to work remarkably well, I have some concerns about the potential for self-correlation to give a false sense of its effectiveness a result of the use of the same data points both for calibration and evaluation of the lidar data.

The entire approach rests on the determination of the relationship between the lidar noise, $B$, and downwelling scattered shortwave radiance for the lidar's (narrow) field of view, $L_s^{\downarrow}$. This is instrument specific and determined by fitting a function to measurements of $B$ and $L_s^{\downarrow}$, ideally – as noted in the paper – this would be done for clear sky conditions. The authors note that there are very limited periods with clear skies in the data set, and they opt instead to use a slightly less direct approach. They use a value of $L_s^{\downarrow}$ estimated for cloudy conditions, with $L_s^{\downarrow}$ derived from a radiative transfer model, STREAMER, with a cloud optical depth estimated from the directly measured total downwelling shortwave irradiance, $F_{SW}^{\downarrow}$. This is shown in figure 4. On the basis of this figure alone, which shows a very strong linear relationship, I think the proposed technique is almost certainly useful. However, when the full technique is implemented: lidar noise is used to estimate $L_s^{\downarrow}$, which is used with the STREAMER results to estimate cloud optical depth and $F_{SW}^{\downarrow}$, the final results are evaluated by comparison with the same N-ICE data used to calibrate the lidar in the first place, and with the same assumptions in place about other quantities that might affect the results: surface albedo, cloud thickness, and cloud microphysical properties. While it is clear that there is a strong relationship between $B$ and $L_s^{\downarrow}$, if some bias is introduced into the calibration because of an invalid assumption – use of a fixed albedo (perhaps too high or low), cloud microphysical properties, cloud thickness – which might affect either the gradient, $K_L$, or offset, $b$, of the linear fit, then that bias CANNOT be identified by the evaluation used here.

The error analysis provided in the appendices deals effectively with random errors – the uncertainty deriving from inherent uncertainty in the measurements or assumed values, but not with any potential mean bias that might arise from the initial calibration.

I appreciate the problem the authors face – they are trying, after the fact, to derive quantities that were never intended to be measured by this instrumentation, and lacking some of the support measurements that one would make if planning this prior to the field campaign. They're done a good job, but could perhaps address the problem of potential calibration bias more effectively.

The available clear sky data is very limited, but it is not zero – May 23 is stated to have 24 hours of clear skies. Do estimates of $B$ and $L_s^{\downarrow}$ from clear sky conditions – however limited – fit the function derived from the cloudy cases?

The period used here, is a 44 days, but only 18 discrete lidar measurements are used, each a 10 minute average, from ~4 measurements per day, so about 10% of the 176 total measurements. While the method clearly has merit, does this very sparse data set imply it's operational use would be likely to return similarly sparse data?

**Detailed comments**

Line 30: the authors state "clouds cover up to 80% of the region at all time and are primarily composed of low-level mixed phase clouds". This true for the summer but not necessarily for the winter season when low level clouds are less frequent.

Line 64: regarding the reference to MOSAiC. Keep an eye on https://online.ucpress.edu/elementa/search-results?fl_SiteID=1000091&page=1&tax=231 – the initial programme overview papers are currently in press, and should be available by the time this paper is accepted.

Line 54: "The surface cloud radiative effect is therefore positive from September to April-May and negative in summer" – it should perhaps be acknowledged that this is also a function of latitude.

Line 159-161: The cloud used for the STREAMER simulations is defined here, and has a fixed altitude and depth. I appreciate that for a given optical depth this probably doesn't greatly affect the results, but I wondered why these weren't height and depth were not taken directly from radiosonde profile – at least for the calibration and to evaluate the range of any impact. Using a better estimate of the actual cloud properties for the calibration of lidar noise against scattered radiance, rather the fixed values, which are then also used for the determination of optical depth from lidar noise, would eliminate one of the closed-loops when validating the method against the same N-ICE data used for the calibration.

Figure 4: It would be useful if on the right hand panel, a line were overplotted showing the location of the peak in $L_s$ as a function of $\theta$ - it would help make clear the ambiguity in $\tau$

Figure 5: the caption states that the results here are plotted for 6 values of $\tau$ (the different coloured points) and 'various values of $\theta$' – I assume that at a given $\tau$ each point corresponds to a different value of $\theta$, but that is not obvious from the figure or caption.

Lines 190-194: this brief description is the closest thing given to a complete description of the method proposed. The various components are covered in more detail in various sections, but it would be useful to give a clear, step by step, breakdown of the full method. Here the description is rather vague, e.g. "…$B$ derived from the lidar helps to determine…"

Line 195: "is simply obtained from the equation detailed by Minnis et al. (1993)." – Minnis et al. (1993) has a lot of equations, none of which perfectly match this one. I think you refer to equation 21, but please cite the intended equation explicitly.

Equation 5: the term $\in_c$ is undefined…until 10 lines below. And the term $c$ – the cloud mask, isn't defined for 5 lines. It would help the reader if all terms were defined immediately after the equation, rather than much later in the discussion.

I'm not sure I understand the full reasoning for equation 5 and its relationship with Minnis eqn 21. Two questions:

1) In the cloudy case, eqn 5 gives the downwelling LW irradiance as the sum of those for the cloud (emitted, from cloud at temperature $T_c$) and the surface (reflected, surface temperature $T_s$). Minnis eqn 21 looks like the same equation:
$F_{LW} = \in B(T_c) + (1-\in)B(T_s)$ , where $B(T) = \sigma T^4$
but it addresses a slightly different situation, the upwelling radiation seen from a satellite, with cloud temperature $T_c$ and $T_s$ the 'clear scene' (surface) temperature. Here the radiation from the surface is seen *through* the cloud rather than being reflected. Why the difference for your case?
2) in the clear case you give the downwelling LW radiation simply as that emitted by the lowest level of the atmosphere (at 2m). I don't understand how that is reasonable, why no contribution from higher levels (or the background of space?) – the atmosphere is largely transparent at the wavelengths concerned here or the measurement of cloud temperature wouldn't be possible.

Line 253: here it is stated that there are 20 points used to calibrate the lidar noise $B$ against the scattered downwelling radiance. I count 18, both on figure 7 and again on figure 6 where the lidar derived cloud optical depth is plotted along with that from N-ICE.

Line 380-381: the authors state that the fact that all LW irradiances were > 230 W m$^{-2}$ suggests that the lidar detected clouds for the entirety of the spring period. Isn't this forced by the fact that the data used to find the irradiances are selected based on the calculated optical depth?

Line 419: 'The skin temperature taken from the ERA5 reanalysis may be colder than the actual skin temperature' – recent evaluation of the ECMWF IFS model in forecast mode, but essentially the same model used to generate ERA5, has a warm bias of ~1K in the skin temperature, at least during the late summer and early autumn. See:

Tjernström, M., G. Svensson, L. Magnusson, I. M. Brooks, J. Prytherch, J. Vüllers, G. Young, 2021: Central Arctic Weather Forecasting: Confronting the ECMWF IFS with observations from the Arctic Ocean 2018 expedition, Quart. J. Roy. Meteorol. Soc. doi:10.1002/qj.3971

**Grammar, typos, etc**

Line 10: 'enables to estimate' -> 'enables us to estimate'

Line 21: 'twice as fast then the rest…' -> 'twice as fast as the rest…'

Line 27: 'contribute to regulate the…' -> 'contribute to the regulation of…'

Line 37: "The underdetermined knowledge on the thermodynamical and radiative feedbacks.." – awkward phrasing, better: "The limited knowledge of the thermodynamical and radiative feedbacks…"

Line 39: "the radiative budget, which is the primary source in the surface energy…" -> "the radiative budget, which is the primary contribution to the surface energy…"

Line 71: 'Buoys have also…' -> 'Buoys also have…'

Line 75: 'fluxes from buoys lidar data' -> 'fluxes from the buoys' lidar data'

Line 80: 'to derive both optical depths and radiative irradiances' -> 'to derive both optical depths and irradiances'

Line 82: 'irradiances measurements at the vicinity of' -> 'irradiance measurements in the vicinity of'

Line 91: 'into the pack ice during several months' -> 'into the pack ice for several months'

Line 92: 'tacked' -> 'tracked'

Line 114: 'at a ice camp' -> 'at an ice camp'

Line 121: 'bandwidths, respectively' -> 'bandwidth, respectively'

Line 129: 'In springtime, Walden et al. (2017); Cohen et al. (2017) indicated' -> 'In springtime, Walden et al. (2017) and Cohen et al. (2017) indicated'

Line 134: 'on 1 minute resolution' -> 'at 1 minute resolution'

Line 137: 'of its emission' -> 'of its emission of LW radiation'

Line 139: 'interpolated at the buoy location' -> 'interpolated to the buoy location'

Line 151: 'same bandwidth as the one of the pyranometer' -> 'same bandwidth as that of the pyranometer'

Line 153: 'by Merkouriadi et al. (2017); Granskog et al. (2018) during' -> 'by Merkouriadi et al. (2017) and Granskog et al. (2018) during'

Line 161: 'water droplets overcoming a 500 m-width cloud layer' -> 'water droplets overlying a 500 m-width cloud layer'

Line 169: 'and decreases afterwards' -> 'and decreases above this value'

Line 238: 'corrected of the Earth-Sun distance' -> 'corrected for the Earth-Sun distance'

Line 279: 'provide a rough information on…' -> 'provide a rough idea of'

Line 287: 'observed the 16 May' -> 'observed on the 16 May'

Line 289: 'whole set N-ICE…' -> 'whole set of N-ICE…'

Line 474: 'A1 Uncertainties on $\tau$' -> 'A1 Uncertainties in $\tau$'

Line 496: 'A2 Uncertainties on…' -> ''A2 Uncertainties in…'

Line 509: 'only depends the uncertainty…' - 'only depends on the uncertainty…'

---

## Author Comment (AC1)

**Dear editor,**

Thank you for your efforts to help with the editorial process with our manuscript. We have organized our response as follows. First, we include the responses to the referees. Then, we include a track changes version of the manuscript with all removed text in red and new text in blue. The goal was to answer each reviewer comment and integrate the related changes into the revised manuscript (attached).

Thanks in advance for your continued work as editor of this manuscript and I look forward to hearing from you in the coming weeks.

**Reviewer #1:**

AC : The authors would like to thank the Reviewer#1 for his/her careful review of our manuscript. We addressed each comment individually and have revised the manuscript accordingly.

The manuscript "Radiative fluxes in the High Arctic region derived from ground-based lidar measurements on board drifting buoys" by Loyer et al. describes a method for calculating broadband radiative SW and LW fluxes from IAOOS buoys featuring lidars. The methodology is built around IAOOS buoys that were co-located with comprehensive surface and upper-air meteorology, and radiometric observations collected during the N-ICE2015 campaign. Pairing the buoys with N-ICE2015 and using the campaign to demonstrate the approach within the environment where the approach will be used is a good experimental design and my feeling is that there is a publishable study here. However, this study, or perhaps this manuscript, is not mature enough for publication in its present form. I hedge here because the first issue that needs to be addressed is the organization and clarity of the text (see below). There are also grammatical errors, awkward phrasing, run on and fragmented sentences, and unnecessary subjective qualifiers (e.g., "rather", "fairly", "mostly", "quite") throughout. It is possible that if the study were better communicated, I would find it more convincing.

A complete rewrite of Sections 2, 3 and much of 4 is needed. There is no logical flow to the narrative as it stands. To begin, explain what steps are needed to calculate fluxes from the buoy, which are currently found spread across all three sections. For example, some necessary introductory information isn't found until deep in Section 4 (Lines 333-337 & 344-349). Explain what information you have from the buoy and what information you need to get from other sources. Maybe make a figure with a flow chart to help readers follow the methodology that begins with lidar backscatter and ends with a flux.

We apologize for the complex structure of the manuscript we had chosen and for confusing wording on our part that lead to some misunderstanding on the objectives of the paper. In particular, we recognize that the structural organization of the methodology was hard to understand as it was split in different sections. As suggested by Reviewer#1, we have completely rearranged Sections 2, 3 and 4 to make them clearer to the readers. Section 2 presents the observational dataset. We made it clearer, with Section 2.1 describing the IAOOS project and measurements available from the buoys and with Section 2.2 introducing observations from the N-ICE expedition. Only IAOOS and N-ICE measurements for spring 2015 (April to June) have been used. The campaign itself and the life duration of the buoy measurements lasted over a longer period, but only a subset of this period has been indeed used due to deposition of frost on

the lidar window. This has been clarified in the text. Section 2.1 and Figure 1 have been modified accordingly.

"This study relies on lidar measurements acquired from April to June 2015 onboard one of the three buoys during this campaign."

The main cause of the confusion in the submitted manuscript was the fact that the approach was spread over two sections intermixed with some of the results. In the new manuscript, we choose a more classical approach : Section 3 presents the methodology and Section 4 the results. As a consequence, the methodology to get the cloud optical depth from, on the one hand, SW flux measurements during the N-ICE expedition and from, on the other hand, the background signal B of the IAOOS lidar, are all included in Sect. 3. This latter also describes the method to obtain the regression slope KI required to convert B to a scattered radiance. Every step of the approach is now presented in Section 3.2 for the SW flux estimation and Sect. 3.3 for the LW flux estimation. All the results of the determination of those variables have been moved to Section 4.1 (previously Section 4.4).

We have also added two schematics (reported below) describing flowcharts to clarify the article structure, the notations used in the paper and to help the readers follow the different steps, which observations are exactly used and where.

Flowchart describing the method to get the downward SW flux  $F\downarrow_{SW}$  from the solar background B measured by the lidar. The estimated optical depth  $\tau_{LAOOS}$ ,  $F\downarrow_{SW}$  and  $F\uparrow_{SW}$  in green are compared to their observed counterparts from N-ICE measurements in red.  $\tau_{LAOOS}$  is then used to estimate the downward LW flux (Fig. 7).

Flowchart describing the method to gt the downward LW flux  $F_{\downarrow IW}$  from the observations onboard the IAOOS buoy (temperature and lidar measurements). Ts and T2m are over this period (April-June 2015) are instead taken from ERA5 (blue) because there are not available from the IAOOS buoy due to instrumental issues (red cross).  $\tau_{IAOOS}$  is obtained from the flowchart describing the retrieval of SW fluxes (Fig. 8).

Even after reading the manuscript, I still don't really understand the purpose of what appears to have been the development of a lookup table using STREAMER in 3.1. I also couldn't understand why so much effort went into trying to make broadband radiometers produce data that is suitable for deriving from lidar (optical depth) because the necessity of doing this was unclear and because there was a micropulse lidar (MPL) deployed on R/V Lance by the US DoE ARM program, which is better suited for comparison. Optical depths were also sometimes calculated using radiosondes instead of being measured by the IAOOS lidar and soundings were used in some cases while reanalysis were used in other cases. By the end of Section 3, I was not even sure how the buoy data was contributing anything of value to the SW flux calculation because Eqs. (3) and (4) were never reconciled.

The development of a lookup table using STREAMER in Sect. 3.1 is used to estimate the downward scattering radiance  $Ls\downarrow$  from the lidar-derived solar background B. Ideally, this would be done under clear sky conditions. Unfortunately there wasn't any clear sky period associated to the IAOOS measurements in 2015. As pointed by Reviewer#2, we have therefore chosen a less direct approach using STREAMER. We have shown a biunivocal relation between the downward scattering irradiances and radiances for each value of the cloud optical depth (Fig. 5). The data derived from the broadband radiometers are first used to estimate the optical depths, leading to a knowledge of Ls values. Those latter are colocalized in time and space with the solar background values derived from the lidar, helping to determinate the value of the lidar constant Kl (Fig. 6 and Eq. 4). This constant Kl is used to convert B into  $Ls\downarrow$  and using STREAMER we can obtain  $F\downarrow_{s,sw}$ .

Backscatter lidar systems as the IAOOS lidar or the micropulse lidar (MPL) deployed on R/V Lance have the ability to 'directly' determine the cloud optical depth, when the top of the cloud can be clearly detected, which is not often the case in the Arctic. Using a comparaison of the signal slopes below and above the cloud, or the Integrated attenuated backscatter, or a Klett or Fernald inversion can lead to an estimate of cloud optical depths. The optical depths directly derived from the lidar signals are not considered as reliable under cloudy skies as the top of the cloud is hardly reached. They are only used in this study to provide a first estimate  $(\tau_0)$  that is compared to the optical depth of the largest scattered radiance, suggesting a range of possible values for the optical depth. A statistics of cloud optical depths 'directly' determined from the IAOOS lidar systems has already been published by Maillard et al. (2021). Here the objective is different: the goal is to present an innovative approach to obtain the shortwave irradiances from the lidar measurements independently. The use of broadband radiometers data is thus necessary for the comparison. In Di Biagio et al. (2020) the optical depth determined from the radiometer data were compared against satellite observations and highlighted a quite similar representation of the optical depth from the radiometer data.

The equations for the LW calculation connected better, but if I understand correctly, the only value the lidar is providing was cloud emissivity.

To estimate the LW fluxes, the lidar data are used both for the determination of the cloud emissivity (function of the optical depth) and the knowledge of the cloud height (driving the effective temperature of the cloud).

The blended use of buoy, ERA5, and N-ICE2015 observations leaves me wondering how you propose to apply this method beyond N-ICE and how the uncertainty will degrade when you don't have N-ICE observations to incorporate. And ultimately/most importantly, does the lidar add enough useful information to beat ERA5 estimates of flux?

The specific use of data from buoys, ERA5 and NICE2015 observations has been clarified in the new version and we hope that the flow charts will make the structure clearer. Observations from NICE2015 radiometers were used in this paper to estimate the Kl constant using a regression analysis. This can be considered as a calibration procedure of the lidar background signals towards the broadband fluxes. This is not expected to vary as a function of the time and could be use to analyze the six years of IAOOS measurements. The ERA5 profiles are used to estimate the molecular transmission and the temperature profiles used in the LW calculation. We could have instead used a model of standard atmosphere (such as the well-known US-Standard Atmosphere 1976), but we believe that the ERA5 results have more chances to be close to reality. Here we also extracted the surface and 2 m temperatures from ERA5 because of issues encountered with the temperature sensors on the buoy used in this study. This information is available on the other IAOOS buoys, enabling to have a determination of the upward longwave fluxes independent of ERA5. An analysis of the six years of IAOOS profiles is very promising as it can provide a statistical database of cloud fraction, cloud altitudes (base and top), optical depths, shortwave and longwave irradiances in a region where those data are rather scarce ; a comparison of ERA5 against those database may highlight potential biases in the reanalyses over sea ice at different seasons and years.

Can you provide some information on the quality of the lidar observations besides the icing issue? There must be all sorts of challenges with level and signal-to-noise, etc.

Mariage et al. (2017) gave a technical description of the IAOOS lidar, the quality of the observations and thoses challenges. The results are based on the first two IAOOS buoys sent in the Arctic Ocean in 2014 and 2015. They have shown that the vertical resolution of the lidar is between 15 m and 60 m. The average solar background calculated in the upper channels (between 25 and 30 km) is subtracted to the signal. Saturation or dead time limitation occurs on the detector in cases of high backscattering signals. The signal is corrected in profiles considered with low or moderate icing. Otherwise, profiles with severe icing are discarded to avoid biases. The revised manuscript has been modified accordingly.

"The lidar system used and its calibration procedure have been fully described by Mariage et al. (2017)"

Lines 53-55: The summertime values do vary and the cycle is not precisely the same everywhere. An extreme example is Greenland, which is positive year-round (https://www.doi.org/10.1175/JCLI-D-15-0076.1)

We thank the reviewer for this comment and this reference. We have mentioned in the new version: "The surface cloud radiative effect in the Svalbard region is positive from September to April-May and negative in summer (Walden et al., 2017; Ebell et al., 2020). Near Greenland, the cloud radiative effect is positive all year round (Miller et al., 2015)"

Lines 56-73: Something is missing here. I understand there are new buoys in the water, but there is no description of what these buoys are measuring.

We have added a description of what these buoys are measuring: "To respond to the need for more observations in the Arctic and compensate the lack of satellite observations at the highest latitudes, multiple buoys have been deployed in the Arctic Ocean between 2014 and 2019. As part of the IAOOS (Ice Atmosphere arctic Ocean Operating System) project (Mariage et al., 2017; Maillard et al., 2021) buoys have been deployed to measure the vertical profile of aerosols or clouds, near surface air temperature, pressure and humidity, snow height and surface temperature as well as ice height and conductivity, temperature, and depth (CTD) measurements "

Section 2.1. Precisely which data is used in the study is unclear. Fig. 1 presents 3 buoys but the caption says only one is used. This contradicts Line 97, which says data from January-June is used, at least "mostly" (??). The end of the paragraph indicates only April-June data is used, and at that only a subset.

We apologize for this confusion. The study indeed only uses a subset of the data and due to icing issues, the winter data were not of sufficient quality to guarantee that the approach developed here gave a good estimate of the radiative fluxes. As a consequence, only the data between April and June have been used. This has been clarified in the revised manuscript. Line 106: "The approach..." is out of place. If this is described later, just remove this sentence.

As the structure has been revised, with the methodology firstly described, this sentence has been removed.

Section 2.2. Some of the information from 2.1 belongs here and not in 2.1.

We agree with this comment and have moved some information from Sect. 2.1 to Sect. 2.2.

*Line 22:* Note that 5 Wm2 and 3% are equivalent. It's more like whichever is larger, 5 Wm2 or 3%, and that 3% is 15 Wm2 at 500 Wm2 SWD.

We agree. Our formulation was unclear and this has been corrected. For the calculation of the uncertainties, we have used the maximum of the errors (max (3%SW, 5W/m2) for example).

Line 124: What is the purpose of this 70% threshold? Why not use all good data that correspond in time to lidar profiles?

The sentence was confusing. We indeed use all good data, which represented 70% of the whole dataset and only 6 days have been removed during the spring period. This has been corrected.

"According to the quality flags (QF) classification introduced by Walden et al. (2017), we only considered observations defined as "good data" (QF=0), which represents 70% of the whole dataset. Finally, we also eliminated 3 days when the IAOOS buoy (IAOOS7) was not yet deployed. This removed a total of 9 days of measurements in spring."

*Line 125: How do you define/did you determine what is "too far"?*

This sentence was not clear. Data from N-ICE that were not co-localized to a IAOOS buoy were discarded. For the spring period, 3 days were removed. This has been reformulated and corrected.

"Finally, three days have been discarded when the IAOOS buoy (IAOOS7) was not deployed yet."

Figure 2: I understand why you use a negative sign for upwelling data in the top two panels, but it is conventional to use all positive signs, and additionally, as plotted the message using the sign and the arrow nomenclature is somewhat redundant and thus cancelling: i.e., isn't a negative of the upward arrow a downward arrow? If you make the plot using positive sign conventions it will be much easier to read.

**We have changed the conventions in Fig. 2 using only positive signs.**

Line 130: By stable do you mean static stability of the atmospheric boundary layer or do you mean that the meteorology did not change much during the period of interest?

**We wanted to say that the meteorology did not change much during the period of interest. We have rewritten the sentence.**

"In springtime, Walden et al. (2017) and Cohen et al. (2017) indicated that the conditions and the surface temperature did not significantly vary. The resulting FLW $\uparrow$  was similar (FLW $\uparrow \simeq 277 \pm 21$  W m-2) throughout the whole period."

Line 131: As a consequence of what?

This has been removed.

*Line 133: Is a sample of clear skies important for some reason?*

The presence of a clear sky period is important to evaluate *STREAMER* fluxes without the presence of clouds. This sentence was moved to Section 4.1 where the derived optical depth values are discussed.

*Line 134: Would you really classify 550 Wm2 on a clear day as extreme?*

We agree with the reviewer that 550 Wm2 on a clear day is likely not an 'extreme' in a meteorological sense but it is the maximum observed during the studied period (called 'extreme' in a mathematical sense). This sentence has been removed for the sake of clarity.

Line 139: Note that this not the native resolution of the ERA5. It is already and interpolated product.

We thank the reviewer for this comment. This has been added in the new version :

"The ERA5 data available at a resolution of 0.25x0.25° is interpolated to the buoy location."

*Line 140: Were the soundings from N-ICE2015 sent to GTS and assimilated by ERA5?*

To our knowledge, the soundings from N-ICE2015 have not been assimilated by ERA5.

Line 150: What is that FOV?

The FOV of the lidar's recever is ~1.4 x 1e6 sr. This has been mentioned in the text. We removed this sentence as the FOV is not set in STREAMER.

Line 200: I don't understand, you are using T2m from ERA5 but earlier you said the buoys measure that.

We have indeed extracted the 2 m temperature from ERA5 because of issues encountered with the temperature sensor on this specific buoy. This information is available on the other IAOOS buoys, enabling to have a determination of the upward longwave fluxes independent of ERA5. This has been clarified in Sect. 2.1 and a figure was added to Sect. 3.3 to describe the consequence of this lack of observations.

Lines 250-252: I don't follow. I feel like you have done a clear sky calculation with STREAMER and are using that to estimate K instead of an observation under clear skies. Is that what you mean?

Lidar observations were not available under clear skies. The Kl constant is estimated under cloudy sky conditions using the data from the NICE 2015 radiometers measurements and STREAMER. To clarify how Kl is determined, more information has been given in Sect. 3.2 :

"Assessing the cloud optical depth from the IAOOS lidar measurements is trickier as it requires a series of steps. The first of them is the calculation of the KL constant. This coefficient is the theoretical slope of the linear dependency of the solar background B on  $L\downarrow$ s, and is a sole function of instrumental and optical properties of the lidar system. An accurate knowledge of the instrumental properties of an autonomous lidar in the Arctic Ocean is however a challenge, not only because there is a limited number of clear-sky days enabling to check the lidar calibration, but also as a frost layer is often deposited on the window of the lidar, disturbing the received signal (Sect. 2.1). We propose here an alternative method to determine the slope KL. The downwelling scattered SW radiance L1s has been calculating using STREAMER with the optical depth  $\tau_{\text{NICE}}$ . This method is applied for each coincident value of B, giving a total of 18 points. Results are shown in Fig. 6. To ensure that the cloud cover is homogeneous in the area observed by the lidar and the radiometers (with different fields of view), a linear regression between B and L1s has been computed only for opaque clouds (Fig. 6). The Pearson correlation coefficient is found to be 0.97. The relation between the measured solar background B and the downwelling scattered SW radiance L1s is written as Eq.4. With KL = 118.54 W-1 m2 sr and b = 3143.53 retrieved from the fitting procedure.The slope KL is applied to the whole dataset sampled aboard the IAOOS buoys. We did not observe any dependency on the solar zenith angle  $\theta$  to build the linear regression. All IAOOS buoys having the same instrumental properties, the same KL could be used for each of them. The uncertainty in KL due to the choice of cloud properties in STREAMER is not significant  $(\Delta KL = 0.38 \text{ W}-1 \text{ m}2 \text{ sr})$ . But the influence of a change in the surface albedo  $\Delta \alpha$  is significant  $(\Delta KL = 5.4 \text{ W}-1 \text{ m}2 \text{ sr})$ . To reduce uncertainties, the albedo obtained from the N-ICE measurements is used to constrain STREAMER runs. The spectral dependency of the solar scattered radiance calculated in the range 200 -3600 nm is implicitly embedded into this slope KL."

Eq. 11: I don't follow. This equation contradicts Eq. 4. If the implicit assumption of Eq. 4 is that b should be 0 then in creating Eq. 11 you have made the assumption that uncertainty in the regression is entirely found in the y-intercept, but it could just as easily be attributed to an error in the slope. Your calculation of K depends on your interpretation of the uncertainty in the regression. If you set b=0 and use the same slope, how much does K change and is that change negligible for the purposes of this study?

We thank the reviewer for this question. Eq.4 was incorrect and has been replaced by Eq.11. The intercept b is always different from zero and is naturally taken into account in the regression analysis.

Line 289: At this stage, I still haven't figured out why we are deriving tau from the N-ICE radiometers. What will this accomplish? If this is simply to be able to characterize the limited number of samples from the buoy within the more continuous time series from N-ICE then some re-organization of the paper is needed to provide a more logical progression of the

steps. I also wonder why you aren't using the micropulse lidar from US DoE ARM, which was installed on R/V Lance? That would make a much better comparison data set for the buoy, wouldn't it, for example avoiding the FOV issues (e.g., Line 315).

**Please see the response to the first comments. The paper has been reorganized to better describe the methodology and the objectives.**

Lines 325-331: I'm a little confused about the effect of the frost. Obviously, if it is attenuating then there is no signal. But if there is signal, is the problem that the partial attenuation from thin frost coverage implicitly propagating into B?

Severe frost (or icing) forming on the receiver's window completely dampens the signal. We used a threshold (called ice index in this study) to only use values that are considered not to have severe icing. A correction is applied to moderate or partial attenuation of the signal from icing. However thin frost coverage is actually implicitly propagating into B. More information on the icing is given in this article: Mariage, V., Pelon, J., Blouzon, F., Victori, S., Geyskens, N., Amarouche, N., Drezen, C., Guillot, A., Calzas, M., Garracio, M., Wegmuller, N., Sennéchael, N., and Provost, C.: IAOOS microlidar-on-buoy development and first atmospheric observations obtained during 2014 and 2015 arctic drifts, Optics Express, 25, A73, https://doi.org/10.1364/oe.25.000a73, 2017

Line 398: I don't understand how upwelling fluxes are coming from IAOOS. The buoy doesn't measure anything relevant for that that you have described. These values are taken from ERA5, interpolated to the position of the buoy?

The reviewer is right. Unfortunately both the albedo and surface temperature are not available on the IAOOS buoy used in this study. In this particular case, the surface temperature has been taken from ERA5. Notwithstanding, surface temperature measurements are available on other IAOOS buoys at different periods and the methodology developed here could be extended to derive the upwelling LW fluxes independently on ERA5 reanalyses.

---

## Author Comment (AC2)

**Dear editor,**

Thank you for your efforts to help with the editorial process with our manuscript. We have organized our response as follows. First, we include the responses to the referees. Then, we include a track changes version of the manuscript with all removed text in red and new text in blue. The goal was to answer each reviewer comment and integrate the related changes into the revised manuscript (attached).

Thanks in advance for your continued work as editor of this manuscript and I look forward to hearing from you in the coming weeks.

**Reviewer #2:**

**AC : The authors would like to thank the Reviewer#2 for his/her careful review of our manuscript. We addressed each comment individually and have revised the manuscript accordingly.**

Overview - This paper documents an initial evaluation of whether the background noise from the lidars installed on IAOOS drifting buoys in the Arctic Ocean can be used to estimate at least some of the components of the surface radiation budget. The approach relies upon combination of the lidar noise measurement combined with information from a radiative transfer model, along with some input from ERA5 reanalysis fields. The method progresses through a number of steps, some of which depend upon assumptions about conditions. The impact of these assumptions is only explored for some of them, and the uncertainties introduced by the others remain unquantified. While well written overall, the method described is complicated, and there are a few places where the discussion becomes confusing and would benefit from being clarified – this often results from a term being used before it is defined – these are noted below. There are also a number of grammatical slip-ups and typos, detailed below.

Major comments - While the approach documented appears to work remarkably well, I have some concerns about the potential for self-correlation to give a false sense of its effectiveness a result of the use of the same data points both for calibration and evaluation of the lidar data. The entire approach rests on the determination of the relationship between the lidar noise, B, and downwelling scattered shortwave radiance for the lidar's (narrow) field of view, *Ls*. *This is instrument specific and determined by fitting a function to measurements of B and*  $Ls\downarrow$ , ideally – as noted in the paper – this would be done for clear sky conditions. The authors note that there are very limited periods with clear skies in the data set, and they opt instead to use a slightly less direct approach. They use a value of Ls testimated for cloudy conditions, with Ls derived from a radiative transfer model, STREAMER, with a cloud optical depth estimated from the directly measured total downwelling shortwave irradiance, FSW1. This is shown in figure 4. On the basis of this figure alone, which shows a very strong linear relationship, I think the proposed technique is almost certainly useful. However, when the full technique is implemented: lidar noise is used to estimate Ls1, which is used with the STREAMER results to estimate cloud optical depth and FSW, the final results are evaluated by comparison with the same N-ICE data used to calibrate the lidar in the first place, and with the same assumptions in place about other quantities that might affect the results:

surface albedo, cloud thickness, and cloud microphysical properties. While it is clear that there is a strong relationship between B and  $Ls\downarrow$ , if some bias is introduced into the calibration because of an invalid assumption – use of a fixed albedo (perhaps too high or low), cloud microphysical properties, cloud thickness – which might affect either the gradient, KL, or offset, b, of the linear fit, then that bias CANNOT be identified by the evaluation used here. The error analysis provided in the appendices deals effectively with random errors – the uncertainty deriving from inherent uncertainty in the measurements or assumed values, but not with any potential mean bias that might arise from the initial calibration.

We thank the reviewer for this detailed useful comment underlying the limits of our approach. We agree with the reviewer, some bias can be introduced into the calibration because of an invalid assumption into STREAMER. We did actually take into account those possible biases and considered assumptions to limit these biases. We calculated the uncertainty on the optical depth, FSW and  $Ls\downarrow$  (shown in appendix) for different assumptions on the values of the albedo, cloud microphysical properties, cloud thickness and cloud height in STREAMER. New Figures are reported below.

Ls $\downarrow$  as a function of COD (left) and FsW (right). The cloud base is set-up at an altitude of 1 km and its thickness is 1 km. The modelled cloud is entirely composed of liquid water droplets of radius 5 µm and  $\varepsilon_s$  is 0.95. The different curves correspond to different albedo ( $\alpha$ ) values: 0.9 (diamonds), 0.5 (circles) and 0.05 (squares). The color scale follows the evolution of the optical thickness.

Ls $\downarrow$  (left) and FSWs $\downarrow$  (right) as a function of optical thickness. The cloud base is set-up at an altitude of 1 km and its thickness is 1 km. The surface parameters are the following:  $\epsilon_s$  is 0.95 and  $\alpha$  is 0.9. The different curves correspond to different hydrometeors :cloud composed of water droplets of radius 5  $\mu$ m (squares), cloud composed of ice crystals as solid columns of radius 5  $\mu$ m (circles), cloud composed of ice crystals as disks of radius 5  $\mu$ m (diamonds). The color scale follows the evolution of the optical thickness.

Ls $\downarrow$  (left) and FSWs $\downarrow$  (right) as a function of optical thickness. The cloud has a thickness of 1 km. The cloud is entirely composed of liquid water droplets of radius 5 µm. The surface parameters are the following:  $\epsilon_s$  is 0.95 and  $\alpha$  is 0.9. The different curves correspond to different cloud base altitudes; 1 km (squares), 5 km (circles), 10 km (diamonds). The color scale follows the evolution of the optical thickness.

A wrong assumption of the albedo would be responsible of large discrepancies on the results of the approach. We have decided to use measured albedo during N-ICE to

constraint STREAMER rather than a fixed one, assuming that the albedo values at the locations of the buoy and the N-ICE ice camp are similar. The induced uncertainty is discussed in Section 4.4. The assumption on the cloud height and thickness have a negligible influence because the optical depth is fixed. The assumption of the cloud microphysical properties may lead to a small bias and it was already taken into account in the uncertainty of the optical depth and FSW.

**We added in the new manuscript the uncertainty on the Kl constant caused by the assumption made in STREAMER, and the new sentences are**

"The uncertainty in KL due to the choice of cloud properties in STREAMER is not significant ( $\Delta$ KL = 0.38 W-1 m2 sr). But the influence of a change in the surface albedo  $\Delta \alpha$  is significant ( $\Delta$ KL = 5.4 W-1 m2 sr). To reduce uncertainties, the albedo obtained from the N-ICE measurements is used to constrain STREAMER runs. The spectral dependency of the solar scattered radiance calculated in the range 200 -3600 nm is implicitly embedded into this slope KL."

I appreciate the problem the authors face – they are trying, after the fact, to derive quantities that were never intended to be measured by this instrumentation, and lacking some of the support measurements that one would make if planning this prior to the field campaign. They're done a good job, but could perhaps address the problem of potential calibration bias more effectively. The available clear sky data is very limited, but it is not zero – May 23 is stated to have 24 hours of clear skies. Do estimates of B and  $Ls\downarrow$  from clear sky conditions – however limited – fit the function derived from the cloudy cases?

**Clear sky conditions are indeed present on May 23, but unfortunately there is no lidar observation at this time. As a consequence, it is not possible to estimate the relation between B and $L_{s\downarrow}$ from clear sky conditions for the studied period.**

The period used here, is a 44 days, but only 18 discrete lidar measurements are used, each a 10 minute average, from ~4 measurements per day, so about 10% of the 176 total measurements. While the method clearly has merit, does this very sparse data set imply it's operational use would be likely to return similarly sparse data?

This is one limitation of our study; in spring, there are not 176 measurements available, but only 65. There are up to 4 measurements a day. Out of those 65 discrete values, only 18 are actually used (28% of the total measurements) because of the icing issues detailed in the paper. The rest has been discarded from the analysis to prevent any potential bias. The heating system onboard the IAOOS buoys has been improved after this campaign; hence a smaller number of lidar observations are biased by icing issues, enabling a more extensive dataset to be used over the six years. In this study however, we are focusing on this sparse dataset, as the comparison against co-localized NICE observations is crucial to assess the performance of the approach.

Line 30: the authors state "clouds cover up to 80% of the region at all time and are primarily composed of low-level mixed phase clouds". This true for the summer but not necessarily for the winter season when low level clouds are less frequent.

We agree with the reviewer that low level clouds are less frequent in winter. We replaced 'all time' by 'spring and summer' in the new manuscript.

*Line* 64: regarding the reference to MOSAiC. Keep an eye on https://online.ucpress.edu/elementa/searchresults?fl\_SiteID=1000091&page=1&tax=231 -the initial programme overview papers are currently in press, and should be available by the time this paper is accepted.

**We thank the reviewer for this updated list of in press papers. The atmospheric reference have been added in the revised manuscript.**

"Shupe, M. D., Rex, M., Blomquist, B., Persson, P. O. G., Schmale, J., Uttal, T., Althausen, D., Angot, H., Archer, S., Bariteau, L., Beck, I., Bilberry, J., Bucci, S., Buck, C., Boyer, M., Brasseur, Z., Brooks, I. M., Calmer, R., Cassano, J., Castro, V., Chu, D., Costa, D., Cox, C. J., Creamean, J., Crewell, S., Dahlke, S., Damm, E., de Boer, G., Deckelmann, H., Dethloff, K., Dütsch, M., Ebell, K., Ehrlich, A., Ellis, J., Engelmann, R., Fong, A. A., Frey, M. M., Gallagher, M. R., Ganzeveld, L., Gradinger, R., Graeser, J., Greenamyer, V., Griesche, H., Griffiths, S., Hamilton, J., Heinemann, G., Helmig, D., Herber, A., Heuzé, C., Hofer, J., Houchens, T., Howard, D., Inoue, J., Jacobi, H.- W., Jaiser, R., Jokinen, T., Jourdan, O., Jozef, G., King, W., Kirchgaessner, A., Klingebiel, M., Krassovski, M., Krumpen, T., Lampert, A., Landing, W., Laurila, T., Lawrence, D., Lonardi, M., Loose, B., Lüpkes, C., Maahn, M., Macke, A., Maslowski, W., Marsay, C., Maturilli, M., Mech, M., Morris, S., Moser, M., Nicolaus, M., Ortega, P., Osborn, J., Pätzold, F., Perovich, D. K., Petäjä, T., Pilz, C., Pirazzini, R., Posman, K., Powers, H., Pratt, K. A., Preußer, A., Quéléver, L., Radenz, M., Rabe, B., Rinke, A., Sachs, T., Schulz, A., Siebert, H., Silva, T., Solomon, A., Sommerfeld, A., Spreen, G., Stephens, M., Stohl, A., Svensson, G., Uin, J., Viegas, J., Voigt, C., von der Gathen, P., Wehner, B., Welker, J. M., Wendisch, M., Werner, M., Xie, Z., and Yue, F.: Overview of the MOSAiC expedition: Atmosphere, Elementa: Science of the Anthropocene, 10, https://doi.org/10.1525/elementa.2021.00060, 2022"

Line 54: "The surface cloud radiative effect is therefore positive from September to April-May and negative in summer" – it should perhaps be acknowledged that this is also a function of latitude.

Our intention was to mention the Svalbard region, where the campaign took place. We agree with the reviewer that this is not the case everywhere in the Arctic. We have modified the text as follows : " The surface cloud radiative effect in the Svalbard region is positive from September to April-May and negative in summer (Walden et al., 2017; Ebell et al., 2020). Near Greenland, the cloud radiative effect is positive all year round (Miller et al., 2015). "

Line 159-161: The cloud used for the STREAMER simulations is defined here, and has a fixed altitude and depth. I appreciate that for a given optical depth this probably doesn't greatly affect the results, but I wondered why these weren't height and depth were not taken directly from radiosonde profile – at least for the calibration and to evaluate the range of any impact. Using a better estimate of the actual cloud properties for the calibration of lidar noise against scattered radiance, rather the fixed values, which are then also used for the determination of optical depth from lidar noise, would eliminate one of the closed-loops when validating the method against the same N-ICE data used for the calibration.

We tested the uncertainty caused by a wrong assumption of the altitude and depth of the cloud in STREAMER on the optical depth, Ls, Fsw and Kl. Little to no change is observed by changing the altitude and depth of the cloud in STREAMER because the

optical depth is given. Instead of using a fixed value of altitude and depth, we are now using the average altitude (400 m) and depth (800 m) determined from the IAOOS lidar measurements. However, it did not lead to a significant change on the results. We have modified the text as :

"Based on the IAOOS lidar's measurements, clouds have a mean geometrical depth with base and top at 400 and 1200 m above mean sea level, respectively. These clouds are assumed to be composed of two cloud layers: a 150 m-width cloud layer composed of 6.9  $\pm$ 1.8 µmdiameter water droplets overlying a 650 m-width cloud layer composed of 25.2  $\pm$ 3.9 µmdiameter hexagonal ice crystals, as described by McFarquhar et al. (2007) in similar conditions."

Figure 4: It would be useful if on the right hand panel, a line were overplotted showing the location of the peak in Ls as a function of  $\theta$ - it would help make clear the ambiguity in  $\tau$ .

**The location of the peak in Ls as a fonction the SZA has been added accordingly in Fig. 4.**

Figure 5: the caption states that the results here are plotted for 6 values of  $\tau$  (the different coloured points) and 'various values of  $\theta' - I$  assume that at a given  $\tau$  each point corresponds to a different value of  $\theta$ , but that is not obvious from the figure or caption.

**This is true. We have mentioned that each point corresponds to a different value of $\theta$ in the caption of figure 5.**

Lines 190-194: this brief description is the closest thing given to a complete description of the method proposed. The various components are covered in more detail in various sections, but it would be useful to give a clear, step by step, breakdown of the full method. Here the description is rather vague, e.g. "...B derived from the lidar helps to determine..."

According to the comments posted by Reviewer#1, we have completely reorganized the paper. The main cause of the confusion in the submitted manuscript was the fact that the approach was spread over two sections intermixed with some of the results. In the new manuscript, we choose a more classical approach : Section 3 presents the methodology and Section 4 the results. As a consequence, the methodology to get the cloud optical depth from, on the one hand, SW flux measurements during the N-ICE expedition and from, on the other hand, the background signal B of the IAOOS lidar, are all included in Sect. 3. This latter also describes the method to obtain the regression slope KI required to convert B to a scattered radiance. Every step of the approach is now presented in Section 3.2 for the SW flux estimation and Sect. 3.3 for the LW flux estimation. All the results of the determination of those variables have been moved to Section 4.1 (previously Section 4.4).

We have also added two schematics (reported below) describing flowcharts to clarify the article structure, the notations used in the paper and to help the readers follow the different steps, which observations are exactly used and where.